# Adaptive Constrained Optimization for Neural Vehicle Routing

## Abstract

Neural solvers have shown remarkable success in tackling Vehicle Routing Problems (VRPs). However, their application to scenarios with complex real-world constraints is still at an early stage. Recent works successfully employ variants of the Lagrange multiplier method to handle such constraints, but their limitation lies in the use of a uniform dual variable across all problem instances, overlooking the fact that the difficulty of satisfying constraints varies significantly across instances. To address this limitation, we propose an instance-level adaptive constrained optimization framework that reformulates the Lagrangian dual problem by assigning each instance its own dual variable. To efficiently optimize this new problem, we design a dual variable-conditioned policy that solves instances with a controllable level of constraint awareness, which effectively decouples policy optimization from the optimization of dual variables. By leveraging this conditioned policy, we customize the optimization of dual variables for each test instance by adapting to its particular constraint violations. Experimental results on the Travelling Salesman Problem with Time Window (TSPTW) and TSP with Draft Limit (TSPDL) show that our method exhibits advantages compared to the strong solver LKH3 and significantly outperforms state-of-the-art neural methods.

## 1. Introduction

The Vehicle routing problem (VRP) is a classic kind of NP-hard combinatorial optimization problem with broad real-world applications in manufacturing (Treitl et al., 2014), transportation (Stein, 1978), and logistics (Konstantakopoulos et al., 2022). VRP solvers in the Operational Research (OR) community, which are typically based on heuris-

[1]Anonymous Institution, Anonymous City, Anonymous Region, Anonymous Country. Correspondence to: Anonymous Author <anon.email@domain.com>.

Preliminary work. Under review by the International Conference on Machine Learning (ICML). Do not distribute.

tic search (Helsgaun, 2000) and integer programming (Applegate et al., 2006), have achieved remarkable success in the past but are often limited by high computational overheads. To address this, neural networks have been leveraged to develop efficient, data-driven heuristics for solving VRPs (Vinyals et al., 2015; Joshi et al., 2019; Kool et al., 2019; Ma et al., 2021; Kim et al., 2021; Jiang et al., 2022; Cappart et al., 2023; Ye et al., 2024; Liu et al., 2024), demonstrating faster solving speeds and competitive solution quality against strong OR solvers. A prominent approach among these neural solvers is utilizing reinforcement learning-based policies to sequentially construct solutions (Bello et al., 2017), which has shown effectiveness on canonical problems like TSP and Capacitated VRP (CVRP) (Kwon et al., 2020; Drakulic et al., 2023; Luo et al., 2023).

Real-world applications of VRP, however, often involve constraints that are more complex than those in the canonical problems. For example, in many business scenarios such as public transportation (Cattaruzza et al., 2017; Shahin et al., 2024) and dial-a-ride systems (Cordeau & Laporte, 2003), the arrival time of vehicle must fall into a customer-requested time window, known as the time window constraint. This constraint significantly restricts the feasible region such that even finding a feasible solution is proved to be NP-complete (Savelsbergh, 1985), which can pose great challenges to most existing solvers. Other examples of complex constraints in VRPs include the global priority rule in disaster relief (Panchamgam, 2011) and the draft limits in maritime transportation (Glomvik Rakke et al., 2012). To handle these hard constraints, classical OR solvers often employ techniques like penalty functions to incorporate constraint violations into the objective function. In the strong solver LKH3 (Helsgaun, 2017), the penalty function is prioritized over the original distance cost, highlighting its emphasis on handling constraints. However, as shown in pervious works (Bi et al., 2024) and our experiments (see Table 1), the feasibility rate obtained by the traditional solvers is still unsatisfactory when runtime budgets are limited.

Neural solvers have achieved remarkable performance on various VRPs, even surpassing LKH3 on large-scale problems (Luo et al., 2024) and specific problem variants (Zheng et al., 2024). However, the research of their extension to VRPs with complex constraints is still at an early stage. To better handle complex constraints, existing studies have re-

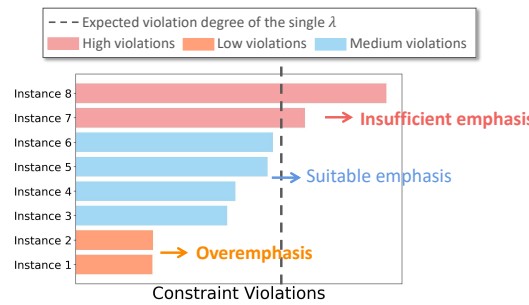

Figure 1. An illustration of the drawback inherent in single-dual variable ($\lambda$) methods. Constraint violations of different problem instances are plotted. The single-$\lambda$ neural solver tends to overemphasize (insufficiently emphasize) constraints on some instances with relatively low (high) constraint violations.

fined neural methods from several perspectives, including constraint-aware feature design (Chen et al., 2024), improvement in network architecture (Falkner & Schmidt-Thieme, 2020), modifications to the objective function (Zhang et al., 2020b; Chen et al., 2022; Tang et al., 2022), and development of novel masking mechanisms (Bi et al., 2024). For instance, Chen et al. (2024) introduced a multi-step look-ahead strategy, integrating the future time window information to enhance constraint-related features. Similarly, Bi et al. (2024) designed a look-ahead-based mask mechanism to proactively exclude actions that would violate constraints in future steps. From the perspective of constrained optimization, Tang et al. (2022) adopted the Lagrange multiplier method to explicitly optimize constraint violations together with the route distance. Notably, the most recent Lagrange multiplier-based method proposed by Bi et al. (2024) has achieved state-of-the-art performance on common benchmarks, regarded as a general and effective solution for complex VRPs. However, these Lagrangian-based methods directly apply the canonical Lagrange multiplier method to neural solvers by using a uniform dual variable across all instances, overlooking the variation in constraint violations among instances, as illustrated in Figure 1. This drawback may severely limit the adaptability of neural models, ultimately leading to performance that is far from optimal.

To address this issue, we introduce a new formulation of the Lagrangian dual problem that assigns each training instance a specific dual variable, enabling adaptive constrained optimization at the instance level. Unlike methods that rely on a single dual variable, this instance-specific formulation offers greater flexibility by optimizing the trade-off between solution quality and constraint satisfaction for each instance. However, directly optimizing the instance-specific dual variables for millions of training instances (e.g., the number of instances is over one hundred millions in the training of POMO (Kwon et al., 2020)) poses significant computational challenges. To mitigate this issue, we develop a dual

variable-conditioned policy, which decouples policy optimization from the optimization of dual variables, thereby reformulating the dual problem into two separate subproblems. First, we focus on solving the inner maximization subproblem by training a dual variable-conditioned policy that is capable of accommodating varying degrees of constraint awareness. This is achieved through a two-stage training strategy: A pre-training stage aimed at fostering adaptability to a wide range of dual variable ($\lambda$) values, and a fine-tuning stage designed to refine the alignment between $\lambda$ values and the hardness of individual instances. Based on the trained $\lambda$-conditioned policy, we solve the outer subproblem in the inference stage by tailoring the optimization of $\lambda$ for each test instance. Through iterative update of $\lambda$, we push the policy to strike an appropriate trade-off between the objective value and constraint violations.

We conduct experiments on two challenging constrained VRPs: Travelling Salesman Problems with Time Window (TSPTW) and TSP with Draft Limit (TSPDL). Notably, these two problems pose greater challenges in satisfying constraints compared to CVRPTW and CVRPDL, as the constraint violations of the latter can be addressed more easily by assigning additional vehicles to the violated nodes. The experimental results demonstrate that our adaptive optimization approach significantly outperforms the state-of-the-art neural method (Bi et al., 2024) that relies on a single dual variable. For example, the performance comparison of optimality gap and infeasibility rate on TSPDL is illustrated in Figure 2. Moveover, compared to the strong solver LKH3 within the same runtime budget, our neural method reduces the infeasibility rate by $95.56\% - 1.33\% = 94.23\%$ on TSPTW100 (i.e., TSPTW with 100 nodes) and $7.02\% - 0.91\% = 6.11\%$ on TSPDL100, while achieving competitive optimality gap. This suggests that neural methods can be a promising direction besides OR solvers for tackling constrained VRPs.

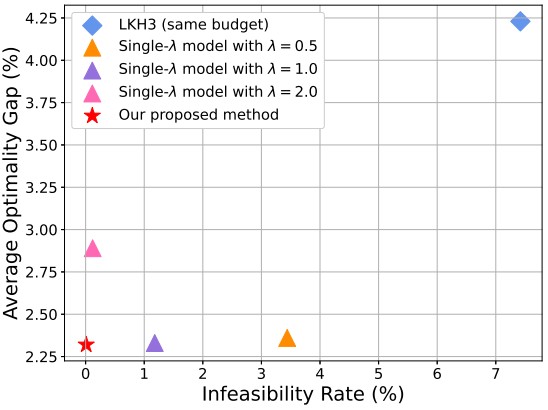

Figure 2. Performance comparison of LKH3, single-$\lambda$ models and our proposed method, on TSPDL with 50 nodes.

## 2. Background

### 2.1. Constrained VRPs

The objective of VRPs (Dantzig & Ramser, 1959) is to determine a tour that minimizes the total travel distance while visiting all the customer nodes. Formally, a VRP instance is defined on a graph $G = (V, E)$, where $V$ represents the set of all customer nodes along with a depot node, $E$ denotes the set of directed edges between each pair of nodes (i.e., the graph is fully connected). The vehicles are required to start and end their tours at the depot node. In this paper, we focus on the TSP with two types of constraints: Time window constraint and draft limit constraint.

**Time window.** The time window constraint nartually arises in many business scenarios that require flexible time scheduling (Toth & Vigo, 2014). In this context, each node is accosiated with a time window $[l_i, u_i]$ that defines the earlist time $l_i$ and the latest time $u_i$ of visiting that node. The time window constraint ensures the arrival time at each node does not exceed the end of its designated time window. If the arrival time $t_i$ is earlier than the start time (i.e., $t_i < l_i$), the vehicle must wait until the time window starts. Formally, a problem instance $I$ is expressed as:

$$\min_{\tau} f_I(\tau) = \sum_{(u,v) \in \tau} d_I(u, v),$$

$$\text{s.t.} \quad g_I(\tau) = \sum_{i=0}^{n-1} \max\{t_i - u_i, 0\} \leq 0,$$

where $\tau$ represents the tour, and $d_I(u, v)$ denotes the travel distance between nodes $u$ and $v$. The goal is to find a tour $\tau$ that minimizes the total distance $f_I(\tau)$ while satisfying the time window constraint $g_I(\tau) \leq 0$.

**Draft limit.** The draft limit in ports is an important factor that influences the routing actions in maritime transportation (Glomvik Rakke et al., 2012). The draft of a ship is the distance between the waterline and the bottom of the ship, affected by the cumulative load. The draft limits in ports are designed to avoid overloaded ships entering these ports. In this context, each node represents a port with a maximum draft $m_i$ and a non-negative demand $\delta_i$. The constraint requires that the cumulative load, $c_i = \sum_{j=1}^{i-1} \delta_{\tau_j}$, over the last $i-1$ steps must not exceed the maximum draft $m_i$ of the $i$-th visited port. Formally, this can be expressed as

$$g_I(\tau) = \sum_{i=0}^{n-1} \max\{c_i - m_i, 0\} \leq 0.$$

### 2.2. Lagrange Multiplier Method

To solve constrained VRPs, the constraint violation can be integrated into the objective function through the formulation of the Lagrangian dual problem (Bertsekas, 2014):

$$\max_{\lambda \geq 0} \min_{\tau} [f_I(\tau) + \lambda \cdot g_I(\tau)],$$

where $\lambda$ is a non-negative dual variable, quantifing the impact of a constraint on the objective function. The Lagrangian dual problem can be optimized by alternatively updating the primal and dual variables. This involves solving the primal problem for a fixed dual variable, which can be addressed using a classical VRP solver, followed by updating the dual variable based on the observed constraint violations (Kohl & Madsen, 1997). The update of the dual variable is often realized using subgradient descent as:

$$\lambda \leftarrow \lambda + \alpha \cdot g_I(\tau),$$

where $\alpha$ is the learning rate. Through the iterative adjustment, the dual variable is continuously refined according to the current level of constraint violation, enabling a better balance between solution quality and constraint satisfaction. More iterative update methods for the dual variable include quadratic method (Hestenes, 1969) and proportional-integral-derivative control (Stooke et al., 2020).

Compared to traditional penalty function-based methods, the Lagrange multiplier method avoids reliance on fixed penalty parameters, providing greater flexibility in handling constraints. Furthermore, it has the potential to yield more optimal solution if the strong duality holds (Boyd & Vandenberghe, 2014). However, the Lagrange multiplier-based method is designed to optimize an individual problem instance. Nartually, a gap arises when it is applied to the training process involving a larger number of instances.

### 2.3. Lagrange Multiplier-based Training Methods for Neural Vehicle Routing

When reinforcement learning (RL) is applied to train neural networks capable of constructing solutions for VRPs (Bello et al., 2017), the expected return of the RL policy $\pi_\theta$ on an instance $I$ is defined as $\mathcal{J}(\pi_\theta, I) = \mathbb{E}_{\tau \sim \pi_\theta(\cdot|I)}[-f_I(\tau)]$, and the expected constraint violation is given by $\mathcal{J}_C(\pi_\theta, I) = \mathbb{E}_{\tau \sim \pi_\theta(\cdot|I)}[-g_I(\tau)]$. Using these definitions, the Lagrangian dual problem of policy optimization is formulated as,

$$\min_{\lambda \geq 0} \max_{\theta} \mathbb{E}_{I \sim D}[\mathcal{J}(\pi_\theta, I) + \lambda \cdot \mathcal{J}_C(\pi_\theta, I)].$$

Unlike typical constrained RL (Achiam et al., 2017; Yao et al., 2023; Gu et al., 2024), where the focus is on solving a specific instance, the trained policy in this framework is designed to generalize to unseen instances from the same problem class. To achieve this, the optimization objective during training involves maximizing the expected performance over a distribution of instances. In practice, the training process is conducted on a dataset $D$ that contains a large number of synthetic problem instances.

To optimize a similar dual problem, Tang et al. (2022) proposed an approach that alternatively updates the policy $\pi_\theta$ and the dual variable $\lambda$. Specifically, the policy $\pi_\theta$ is optimized by policy gradient algorithms such as RE-INFORCE (Williams, 1992), while the dual variable $\lambda$ is optimized by subgradient descent. This method balances the trade-off between minimizing the objective and reducing constraint violations by dynamically adjusting $\lambda$. More recently, Bi et al. (2024) observed that optimizing $\lambda$ may incur significant computational overhead due to the additional iterations required for updating $\lambda$. Therefore, they fixed $\lambda$ to a pre-defined constant throughout the training process.

**Limitations of Lagrangian-based training.** The Lagrange multiplier method was originally designed for optimizing a single instance. However, existing approaches directly extend this method to the training process of neural solvers by utilizing a single shared dual variable for a large number of training instances. This simplification overlooks the fact that different instances can exhibit significantly varying levels of constraint violations, as demonstrated in Figure 1, thereby resulting in suboptimal performance.

## 3. Method

To address the aforementioned limitations, we propose an Instance-level adaptive Constrained Optimization (ICO) method. A graphical illustration of our method is shown in Figure 3. In this section, we first provide an overview of the proposed ICO approach, followed by a detailed description of its training process and network architecture.

### 3.1. Instance-level Adaptive Constrained Optimization

We leverage instance-specific dual variables to effectively handle the varying degrees of constraint violations across instances, which can enable a more flexible trade-off between optimizing the objective and satisfying the constraints. Formally, the new dual problem is formulated as

$$\min_{\{\lambda_i\}_{i=1}^{N}} \max_{\theta} \sum_{i=1}^{N} [\mathcal{J}(\pi_\theta, I_i) + \lambda_i \cdot \mathcal{J}_C(\pi_\theta, I_i)], \quad (1)$$

where $N$ is the number of training instances and $\lambda_i$ is the dual variable specific to instance $I_i$. This dual formulation potentially leads to enhanced performance in both solution quality and constraint satisfaction if the prime and dual variables are both optimized properly. However, it is extermely challenging and computationally expensive to optimize the instance-specific dual variables for **millions of training instances**. In the common training method of neural solvers (Kwon et al., 2020), more than one hundred million training instances are generated on the fly, and each instance is only used once during training without additional iterations to refine its corresponding dual variable.

This training process necessitates an efficient and scalable approach to adaptively manage instance-specific dual variables. Therefore, we discard the expensive iterative method and decouple the original bi-level optimization problem into two separate subproblems: Solve the inner subproblem of Eq. (1) as phase 1 and solve the outer subproblem based on the inner results as phase 2.

**Phase 1: Solve the inner subproblem.** In the first phase, we solve the inner maximization problem separately while considering varying values of $\lambda$, aiming to obtain a manifold of policies capable of solving instances with continuously varying levels of constraint awareness. To achieve this, we propose training a $\lambda$-conditioned policy $\pi_\theta(\cdot|\lambda)$ that takes $\lambda$ as input and performs as trained using the specified $\lambda$, i.e.,

$$\pi_\theta(\cdot|\lambda) \approx \arg\max_{\pi} \sum_{i=1}^{N} [\mathcal{J}(\pi, I_i) + \lambda \cdot \mathcal{J}_C(\pi, I_i)],$$

where the right side represents the optimal policy corresponding to the given $\lambda$. With this condition mechanism, the constraint sensitivity of the policy can be seamlessly controlled by adjusting the input value of $\lambda$, without requiring any modification to the network parameters. This can effectively decouple the policy optimization process from the optimization of the dual variables, thereby enhancing scalability of the Lagrangian-based training method. The detailed training algorithm and network architecture for the $\lambda$-conditioned policy are provided in Section 3.2.

**Phase 2: Solve the outer subproblem.** The second phase is performed during the inference stage, where instance-specific $\lambda$ values are optimized based on the feedback provided by the trained $\lambda$-conditioned policy. For each new instance, we iteratively update $\lambda$ by subgradient descent to minimize its specific constraint violations, thereby adjusting the policy to achieve an appropriate trade-off. This process alternates between sampling a solution using the policy $\pi_\theta(\cdot|\lambda)$ and updating $\lambda$ based on the observed constraint violations of the sampled solution. Formally, the process is described as follows:

$$\tau_{t-1} \sim \pi_\theta(\cdot|\lambda_{t-1}, I), \quad \lambda_t = \lambda_{t-1} + \alpha \cdot g_I(\tau_{t-1}),$$

where $t$ denotes the iteration timestep, and $g_I(\tau_{t-1})$ is the constraint violation of the sampled solution. Note that we initialize all $\lambda$ values using an identical $\lambda_0$. Furthermore, we also explore to utilize Proportional-Integral-Derivative (PID) control to adjust the $\lambda$-value as suggested by Stooke et al. (2020), detailed in Appendix E.2.

### 3.2. Dual Variable-Conditioned Policy

The $\lambda$-conditioned policy serves as a key component in optimizing the decoupled dual problem. We design a two-stage

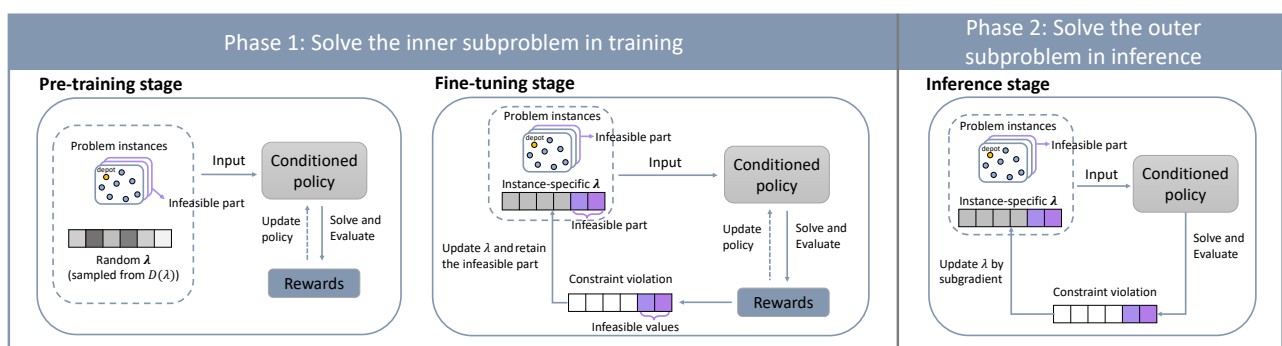

*Figure 3.* An illustration overview of the proposed method. The bi-level Lagrangian problem for constrained optimization is decoupled into two phases. Left (phase 1): Solve the inner subproblem in training. Right (phase 2): Solve the outer subproblem in inference.

training algorithm for the $\lambda$-conditioned policy, consisting of a pre-training stage to develop adaptability to diverse $\lambda$ values and a fine-tuning stage to achieve a more precise alignment between $\lambda$ values and instance hardness. Detailed description of the two training stages is as follows.

**Pre-training stage.** The pre-training stage is conducted on randomly sampled $\lambda$ values, thereby enabling the model to generalize effectively across varying $\lambda$ conditions. The training objective can be expressed as

$$\max_\theta \mathbb{E}_{I\sim D}\mathbb{E}_{\lambda\sim D_\lambda}[\mathcal{J}(\pi_\theta(\cdot|\lambda), I) + \lambda\mathcal{J}_C(\pi_\theta(\cdot|\lambda), I)].$$

Specifically, we randomly sample $\lambda_i$ from a pre-defined distribution $D_\lambda$ for each training instance $I_i$, constituting a pair sample $(\lambda_i, I_i)$. The reward function of the instance $I_i$ is reweighted by its own dual variable $\lambda_i$. Following the shared baseline method (Kwon et al., 2020), we sample multiple solutions $\{\tau^j\}_{j=1}^P$ for each $(\lambda_i, I_i)$ pair and estimate the baseline by the average reward of these solutions. Then, we compute the policy gradient $\nabla_\theta J(\theta)$ using the REINFORCE (Williams, 1992) algorithm as

$$R^j = -(f_{I_i}(\tau^j) + \lambda_i(g_{I_i}(\tau^j) + c_{I_i}(\tau^j))), \forall j \in [P],$$

$$\nabla_\theta J(\theta) = \frac{1}{P}\sum_{j=1}^{P}(R^j - \frac{1}{P}\sum_{k=1}^{P}R^k)\log\pi_\theta(\tau^j|\lambda_i, I_i),$$

where $[P]$ denotes $\{1, ..., P\}$, and $c_{I_i}(\tau^j)$ is the number of timeout nodes, which we use as a heuristic penalty reward, following the reward design of (Bi et al., 2024). The factor $R^j - \frac{1}{P}\sum_{k=1}^{P}R^k$ represents the *advantage* that measures relative reward improvement over the shared baseline. Intuitively, the training algorithm reinforces the probability of generating positive advantage trajectories (i.e., solutions) while decreasing the probability of generating negative ones. Through this training process with random $\lambda$, the conditioned policy obtains the adaptability to different levels of constraint awareness. The pseudo code of the pre-training process is provided in Appendix A.

**Fine-tuning stage.** To achieve an effective alignment between $\lambda$ values and instance hardness, we further fine-tune the pre-trained policy using iteratively updated $\lambda$ values. In this stage, we initialize a uniform and small initial value $\lambda^{(0)}$ for all instances and alternate between optimizing the policy and updating the dual variables strictly following the original formulation in Eq. (1). For policy optimization, we continue to employ the REINFORCE algorithm with an average baseline, as used in the pre-training stage. For updating the dual variables, the subgradient is computed based on the minimal constraint violation value across a set of sampled solutions $\{\tau^j\}_{j=1}^P$. Formally, the $\lambda$ values are updated by the following rule:

$$\lambda_i^{(t)} = \lambda_i^{(t-1)} + \alpha \min_{j\in[P]}(g_{I_i}(\tau^j) + c_{I_i}(\tau^j)),$$

where $\alpha$ is the learning rate. After each iteration, we retain the infeasible instances and their corresponding $\lambda$ values in the batch while replacing the feasible instances with new ones. It is important to note that the pre-trained policy is already capable of finding feasible solutions for the majority of instances. Therefore, the proportion of infeasible instances in each batch is typically small, ensuring that the iterations for updating $\lambda$ do not significantly affect computational efficiency. Moreover, to further enhance training efficiency and avoid excessive focus on particularly hard instances, we impose a maximum iteration limit and a cap on the infeasible instance ratio. The pseudo code of the fine-tuning process is provided in Appendix A.

**Network architecture.** The $\lambda$-conditioned policy solves instances with a controllable level of constraint awareness, determined by the condition variable $\lambda$. Similar conditioned policies have been explored in related works, particularly for multi-objective optimization (Lin et al., 2022; Wang et al., 2024) and latent space search (Chalumeau et al., 2023). Among them, there are two possible ways to incorporate the target variable into the policy network: (1) embedding it into the initial input features or (2) embedding it into the decoder's context. In this paper, we adopt the

$\lambda$-conditioned initial embedding, which empirically demonstrates superior performance in adjusting trade-off behaviors (see Appendix E.3). Specifically, building on the POMO model (Kwon et al., 2020), we incorporate a linear transformation of $\lambda$ into the original initial embeddings. The embedding is computed as:

$$\boldsymbol{h}_i^{(0)} = W^\lambda \lambda + W^h [x_i, y_i, l_i, u_i]^\top,$$

where $W^\lambda \in \mathbb{R}^{d \times 1}$ and $W^h \in \mathbb{R}^{d \times 4}$ are trainable parameters, and $[x_i, y_i, l_i, u_i]$ represents the concatenation of the node's coordinates $(x_i, y_i)$ and its time window bounds $(l_i, u_i)$. This concatenated feature vector serves as the input representation for each node. The output $\boldsymbol{h}_i^{(0)}$ is then used as the initial embedding for the encoder network, which employs the multi-head attention mechanism (Vaswani et al., 2017) to perform message passing and update node embeddings. Intuitively, the $\lambda$-conditioned embedding adjusts the relative importance of distance-related features (e.g., node coordinates) and constraint-related features (e.g., time window bounds) based on the value of $\lambda$, thereby enabling a controllable level of constraint awareness. The rest of the network architecture closely follows the standard POMO model (Kwon et al., 2020).

## 4. Experiments

In this section, we evaluate the effectiveness of our ICO method through comparison experiments and ablation studies. The key questions that our experiments will address are as follows: (1) Does our proposed method outperform single-$\lambda$ models trained with different $\lambda$ values? (2) What advantages can our neural method offer compared to strong OR solvers, such as LKH3 (Helsgaun, 2017)?

### 4.1. Experimental Settings

**Problem instance generation.** We conduct our experiments on two kinds of problems: TSPTW and TSPDL. Following prior works (Kool et al., 2019), we randomly sample node coordinates $(x_i, y_i)$ from a uniform distribution $U(0, 1)$ within a square. For generating the time windows and draft limits, we utilize the code of Bi et al. (2024) and adopt the **hard** settings, which are sufficiently challenging to examine state-of-the-art neural and OR solvers.

**Implementation details.** Our model is implemented based on the POMO framework (Kwon et al., 2020), incorporating the PI mask (Bi et al., 2024) to restrict the search space. We only employ the PIP decoder to predict masks during the training process on TSP instances with $n = 100$. The prior distribution of $\lambda$ in the pre-training stage, i.e., $D(\lambda)$, is set to a triangular distribution $T(0.1, 0.5, 2.0)$. The learning rate for updating $\lambda$ is set to 0.5 for TSPTW and 0.2 for TSPDL. The common hyperparameters shared between our

method and prior works follow their default settings (Kwon et al., 2020; Bi et al., 2024). More implementation details are provided in Appendix D due to space limitation.

**Baselines.** We compare our proposed method against state-of-the-art neural methods and OR solvers. For OR solvers, we include LKH3 (Helsgaun, 2017), one of the strongest solver specifically designed for VRPs; OR-Tools (Falkner & Schmidt-Thieme), a general-purpose solver capable of handling various constraints; and two greedy heuristics, Greedy-L and Greedy-C. Greedy-L selects the nearest node at each step, while Greedy-C chooses the node with the shortest remaining time for TSPTW (or the minimal draft limit for TSPDL). For neural methods, we consider the state-of-the-art approaches: AM+PIP and POMO+PIP (Bi et al., 2024). For TSPTW100 and TSPDL100, we report the results of the models trained with the PIP decoder. Note that the POMO+PIP model can be considered as the single-$\lambda$ policy, serving as a clear ablation of our adpative method.

**Metrics.** We evaluate performance and efficiency using four metrics: infeasibility rate, average optimality gap, normalized HyperVolumn (HV) and runtime. Among these, the HV serves as a comprehensive indicator, capturing both feasibility and solution quality. A detailed explanation of these metrics is provided in Appendix D.3.

**Evalution configurations.** Our method employs $\times 8$ instance augmentation and 16 iterations to update $\lambda$ during the inference stage. To align the runtime consumption, we use *sampling* inference strategy for POMO+PIP and AM+PIP. Detailed configurations is provided in Appendix D.3.

### 4.2. Main Results

**Comparison with single-$\lambda$ models.** The performance comparisons on TSPTW and TSPDL across different problem scales are presented in Table 1. On TSPTW100, the proposed ICO method reduces the infeasibility rate from $4.33\%$ (achieved by POMO+PIP with $\lambda = 1.0$) to an impressive $1.33\%$, representing a substantial reduction of $3.00\%$. Similarly, on TSPTW50, the infeasibility rate is lowered from $1.56\%$ to just $0.51\%$. Even when the $\lambda$ value in single-$\lambda$ models is increased to 2.0, these models still lags behind the ICO method in terms of feasibility, with the sole exception being TSPDL100. In addition to improving feasibility rates, the ICO method consistently outperforms single-$\lambda$ models in terms of optimality gaps. For instance, the ICO method achieves a smaller gap of $9.22\%$ on TSPDL100, compared to $10.77\%$ achieved by the best POMO+PIP model. Moreover, the ICO method showcases the highest HV scores on all benchmarks, further highlighting its comprehensive performance on both feasibility and solution quality. For example, on TSPDL100, the HV improves significantly from

*Table 1.* Experimental results on TSPTW and TSPDL. Test instances are generated using the hard settings. The results of Greedy-L, Greedy-C, full time LKH3 and OR-Tools are drawn from exisiting papers (Bi et al., 2024). LKH3 (less time) and OR-Tools (less time) denote the OR methods with reduced runtime budgets to align with neural solvers. For AM+PIP and POMO+PIP, we report the results obtained by using the *sampling* inference. The best and the runner-up results are highlighted in Blue and Violet, respectively.

| Methods | TSPTW ($n = 50$) | | | | TSPTW ($n = 100$) | | | |
|---|---|---|---|---|---|---|---|---|
| | Inf. Rate ↓ | Avg. Gap ↓ | HV ↑ | Time ↓ | Inf. Rate ↓ | Avg. Gap ↓ | HV ↑ | Time ↓ |
| LKH3 | 0.12% | 0.0% | 1.00 | 7h | 0.07% | 0.0% | 1.00 | 1.4d |
| OR-Tools | 65.72% | 0.0% | 0.34 | 2.4h | 89.07% | 0.0% | 0.11 | 1.6d |
| Greedy-L | 100.0% | / | / | 21.8s | 100.0% | / | / | 1.3m |
| Greedy-C | 72.55% | 1.53% | 0.19 | 4.5s | 93.38% | 1.43% | 0.05 | 11.1s |
| LKH3 (less time) | 57.34% | 0.01% | 0.43 | 100s | 95.56% | 0.03% | 0.04 | 8m |
| OR-Tools (less time) | 65.72% | 0.02% | 0.34 | 99s | 89.07% | 0.51% | 0.10 | 8m |
| AM + PIP ($\lambda = 1.0$) | 2.99% | 0.34% | 0.90 | 105s | 7.80% | 0.70% | 0.79 | 8m |
| POMO + PIP ($\lambda = 0.5$) | 1.95% | 0.08% | 0.96 | 108s | 4.90% | 0.17% | 0.92 | 9m |
| POMO + PIP ($\lambda = 1.0$) | 1.56% | 0.16% | 0.95 | 108s | 4.33% | 0.25% | 0.91 | 9m |
| POMO + PIP ($\lambda = 2.0$) | 1.41% | 0.19% | 0.95 | 108s | 4.71% | 0.39% | 0.88 | 9m |
| ICO (Ours) | 0.51% | 0.07% | 0.98 | 91s | 1.33% | 0.14% | 0.96 | 8m |

| Methods | TSPDL ($n = 50$) | | | | TSPDL ($n = 100$) | | | |
|---|---|---|---|---|---|---|---|---|
| | Inf. Rate ↓ | Avg. Gap ↓ | HV ↑ | Time ↓ | Inf. Rate ↓ | Avg. Gap ↓ | HV ↑ | Time ↓ |
| LKH3 | 0.0% | 0.0% | 1.00 | 6.8h | 0.0% | 0.0% | 1.00 | 1.2d |
| OR-Tools | 100.0% | / | / | 10.6s | 100.0% | / | / | 56.8s |
| Greedy-L | 100.0% | / | / | 2.4m | 100.0% | / | / | 9.4m |
| Greedy-C | 0.0% | 99.73% | / | 10.9s | 0.0% | 156.37% | / | 25s |
| LKH3 (less time) | 7.42% | 4.23% | 0.20 | 70s | 7.02% | 6.76% | 0.20 | 6m |
| OR-Tools (less time) | 100.0% | / | / | 3s | 100.0% | / | / | 29s |
| POMO + PIP ($\lambda = 0.5$) | 3.44% | 2.36% | 0.58 | 71s | 62.94% | 20.95% | / | 5m |
| POMO + PIP ($\lambda = 1.0$) | 1.18% | 2.33% | 0.78 | 71s | 3.23% | 10.77% | 0.31 | 5m |
| POMO + PIP ($\lambda = 2.0$) | 0.12% | 2.89% | 0.85 | 71s | 0.11% | 12.24% | 0.38 | 5m |
| ICO (Ours) | 0.01% | 2.32% | 0.88 | 69s | 0.91% | 9.22% | 0.49 | 5m |

0.38 to 0.49, while on TSPTW100, it increases from 0.92 to 0.96. These results demonstrate that the ICO method is capable of generating high-quality solutions while maintaining a higher level of feasibility. For further comparisons, we also present the anytime performance in Appendix E.5 .

**Comparion with strong OR solvers.** In Table 1, we also compare our neural methods with strong OR solvers, LKH3 and OR-Tools, under aligned runtime conditions. The results show that our ICO method achieves a dramatic improvement in infeasibility rates, reducing them from 95.56% to 1.33% (a 94.23% reduction) on TSPTW100 and from 7.02% to 0.91% (a 6.11% reduction) on TSPDL100. Regarding solution quality, our method consistently outperforms OR-Tools across all benchmarks and even surpasses LKH3 on TSPDL50 in terms of average gap. While the solution quality of our neural approach on the other three benchmarks still lags behind LKH3, the substantial improve-

ments in feasibility and competitive performance overall underscore the strengths of neural methods compared to strong OR solvers. Note that the full time LKH3 still achieves the best performance among all methods in terms of both infeasibility rate and average gap; however, its runtime is extermely long, even exceeding an entire day on TSPTW100 and TSPDL100. Additionally, it is observed that the Greedy-C algorithm obtains near-zero infeasibility rates on TSPDL, but its average gaps remain significantly poor.

### 4.3. Additional Study

In this subsection, we present a series of experiments to investigate the impact of training stages, the update rules for $\lambda$, and different network architectures. Other analyses regarding the distribution $D(\lambda)$ and the anytime performance, are provided in Appendix E due to space limitation.

**Analysis of the training stages.** Figure 4 illustrates the performance of POMO+PIP (with $\lambda = 1$), the pre-trained policy, and the fine-tuned policy. The comparison between the pre-trained and fine-tuned policies reveals that the fine-tuning process leads to a substantial reduction in both infeasibility rate and average gap, thereby demonstrating its effectiveness in enhancing model performance. Notably, even the pre-trained policy alone surpasses the single-$\lambda$ model (POMO+PIP), further highlighting the advantages of the proposed training approach. Results on other benchmarks are detailed in Appendix E.1 due to space limitation.

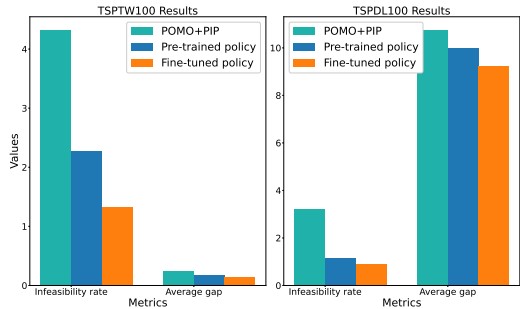

*Figure 4.* Comparison of the pre-trained policy and fine-tuned policy on TSPTW100 and TSPDL100.

**Analysis of update rules for $\lambda$ in inference stage.** In Appendix E.2, we evaluate the performance of the proposed ICO method under different strategies for updating $\lambda$ values in inference stage: fixed $\lambda$ values (0.5, 1.0, and 2.0), randomly sampled $\lambda$ values, the subgradient descent method and the PID control method (Stooke et al., 2020). For the random sampling strategy, $\lambda$ values are drawn randomly from the uniform distribution $U(0.1, 2.0)$ at each iteration. The results in the last three rows indicate that both the subgradient descent method and the PID control method generally outperform the random sampling strategy, with particularly improvements in reducing the infeasibility rate. Notably, it is observed that the random sampling is also a performant strategy, demonstrating that just randomly varying $\lambda$ values for each instance is effective. Moreover, as evidenced in the first three rows, employing fixed $\lambda$ values leads to significantly inferior performance compared to the adaptive variation of $\lambda$, underscoring the critical importance of dynamically adjusting $\lambda$ for each instance.

**Analysis of the network architecture.** In Appendix E.3, we compare the performance of the network with conditioned context and network with conditioned embeddings. The experimental results demonstrate that the conditioned embedding method achieves significantly superior performance in both infeasibility rate and average optimality gap. This performance advantage can be attributed to the fact that the conditioned embedding method can utilize the capacity of the entire network to process $\lambda$-related information.

## 5. Related works

**Prevalent paradigms of neural VRP.** Many researchers have focused on end-to-end neural methods that learn to generate solutions through deep neural networks (Bengio et al., 2021; Cappart et al., 2023). These neural solvers can be categorized into three paradigms (Ma et al., 2023): learn-to-construct methods (Nazari et al., 2018), learn-to-predict methods (Joshi et al., 2019; Sun & Yang, 2023) and learn-to-search methods (Ma et al., 2021). Detailed introduction of these paradigms are in Appendix B due to space limitation.

**Recent advances in neural VRP.** Recent advancements in neural methods for solving VRPs focus on improving scalability (Fu et al., 2021; Luo et al., 2023; Ye et al., 2024; Gao et al., 2024a; Fang et al., 2024) and robustness (Jiang et al., 2022; Bi et al., 2022; Zhou et al., 2023; Jiang et al., 2023) through innovative architectures and learning strategies. Detailed description of these related works are provided in Appendix B due to space limitation. Besides these efforts, this paper focuses on complex constrained VRPs, which are common in real-world applications (Cattaruzza et al., 2017; Glomvik Rakke et al., 2012) but have not received much attention in the research community. Only a few works (Tang et al., 2022; Chen et al., 2024; Bi et al., 2024) try to address it through feature enhancement or Lagrange multiplier method. In this context, we introduce a novel instance-level adpative framework for Lagrangian-based neural methods, reducing the infeasiblity rate significantly.

## 6. Conclusion

In this paper, we propose a novel approach ICO to address the limitations of existing Lagrangian-based neural methods in solving complex constained VRPs. Unlike prior methods that rely on a single, uniform dual variable across all problem instances, ICO leverages instance-specific dual variables to improve adaptability and optimize the trade-off between solution quality and constraint satisfaction for every problem instance. Experimental results on two challenging constrained VRP benchmarks, TSPTW and TSPDL, demonstrate that ICO significantly reduces infeasibility rates compared to both state-of-the-art neural methods and strong OR solvers like LKH3, while achieving competitive or improved solution quality under aligned runtime budgets. These empirical findings suggest that our ICO framework can be a promising alternative for strong OR solvers when tackling constrained combinatorial problems. Notably, the ICO framework is not confined to the VRP domain but can be extended to other areas such as scheduling, packing, and general constrained optimization. Future works could focus on predicting optimal $\lambda$ values based on instance features, refining training strategies of the conditioned policy, and enabling the generalization ability across various constraints.

## Impact Statement

This paper presents work whose goal is to advance the field of Machine Learning. There are many potential societal consequences of our work, none which we feel must be specifically highlighted here.

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

# A. Pseudo Code of the Training Process

---

**Algorithm 1** Pre-training of the $\lambda$-conditioned policy

---

**Input:** Distribution $D_\lambda$, number of training batches $T$, batch size $B$, number of parallel sampling $P$

Initialize policy network parameters $\theta$

**for** $t = 0$ **to** $T - 1$ **do**

    Generate a batch of instances $\{I_i\}_{i=1}^B$

    Sample dual variables $\lambda_i \sim D_\lambda, \quad \forall i \in \{1, ..., B\}$

    Sample multiple solutions $\{\tau_i^j\}_{j=1}^P \sim \pi_\theta(\cdot|\lambda_i, I_i), \quad \forall i \in \{1, ..., B\}$

    Compute baseline $b_i \leftarrow \frac{1}{P} \sum_{j=1}^P -(f_{I_i}(\tau_i^j) + \lambda_i g_{I_i}(\tau_i^j)), \quad \forall i \in \{1, ..., B\}$

    Compute policy gradient $\nabla_\theta J(\theta) \leftarrow \frac{1}{BP} \sum_{i=1}^B \sum_{j=1}^P (-(f_{I_i}(\tau_i^j) + \lambda_i g_{I_i}(\tau_i^j)) - b_i) \nabla_\theta \log \pi_\theta(\tau_i^j|\lambda_i, I_i)$

    Update parameters $\theta \leftarrow \theta + \alpha \nabla_\theta J(\theta)$

**end for**

**Output:** $\theta$

---

---

**Algorithm 2** Fine-tuning of the $\lambda$-conditioned policy

---

**Input:** Number of training batches $T$, batch size $B$, number of parallel sampling $P$, dual variable learning rate $\alpha_\lambda$, policy learning rate $\alpha$, maximum number of itertations $K$, maximum infeasible ratio $\delta$

Initialize policy network parameters $\theta$

Generate a batch of instances $\{I_i\}_{i=1}^B$

Initialize dual variables $\lambda_i \leftarrow \lambda^{(0)}, \quad \forall i \in \{1, ..., B\}$

Initialize iteration counts $k_i \leftarrow 0, \quad \forall i \in \{1, ..., B\}$

**for** $t = 0$ **to** $T - 1$ **do**

    Sample multiple solutions $\{\tau_i^j\}_{j=1}^P \sim \pi_\theta(\cdot|\lambda_i, I_i), \quad \forall i \in \{1, ..., B\}$

    Compute baseline $b_i \leftarrow \frac{1}{P} \sum_{j=1}^P -(f_{I_i}(\tau_i^j) + \lambda_i g_{I_i}(\tau_i^j)), \quad \forall i \in \{1, ..., B\}$

    Compute policy gradient $\nabla_\theta J(\theta) \leftarrow \frac{1}{BP} \sum_{i=1}^B \sum_{j=1}^P (-(f_{I_i}(\tau_i^j) + \lambda_i g_{I_i}(\tau_i^j)) - b_i) \nabla_\theta \log \pi_\theta(\tau_i^j|\lambda_i, I_i)$

    Update parameters $\theta \leftarrow \theta + \alpha \nabla_\theta J(\theta)$

    Adjust the maximum number of iterations $K$ according to the current infeasibility ratio, ensuring the ratio of retained infeasible instances does not exceed the maximum ratio $\delta$

    **for** each instance $I_j$ without feasible solutions **do**

        Update $\lambda_j \leftarrow \lambda_j + \alpha_\lambda \min_{m \in [P]} (g_{I_j}(\tau_j^m) + c_{I_j}(\tau_j^m))$

        Increment $k_j \leftarrow k_j + 1$

    **end for**

    **for** each instance $I_j$ with zero $k_j$ or $k_j > K$ **do**

        Generate a new instance to replace $I_j$

        Initialize $\lambda_j \leftarrow \lambda^{(0)}$ and $k_j \leftarrow 0$

    **end for**

**end for**

**Output:** $\theta$

---

# B. Related Works

**Prevalent paradigms of neural VRP.** Many researchers have focused on end-to-end neural methods that learn to generate solutions through deep neural networks (Bengio et al., 2021; Cappart et al., 2023). These neural solvers can be categorized into three paradigms (Ma et al., 2023): (1) **Learn-to-Construct (L2C) methods** sequentially extends solutions from scratch in an autoregressive manner, typically trained via reinforcement learning (Nazari et al., 2018) or imitation learning (Drakulic et al., 2023). These L2C methods have proven to be applicable to a variety of combinatorial problems (Zhang et al., 2020a) and industrial applications (Lai et al., 2022). (2) **Learn-to-Predict (L2P) methods** operate under a variable-independent assumption, directly predicting the entire solution without conditional dependence (Joshi et al., 2019). While computationally efficient, L2P methods often suffer from limited expressiveness. To address this issue, recent research has introduced diffusion models to enhance the L2P paradigm by leveraging their ability to generate multimodal distributions of optimal

solutions (Sun & Yang, 2023; Li et al., 2023). (3) **Learn-to-Search (L2S) methods** adopt the iterative framework of traditional search heuristics. During the search process, L2S methods usually leverage a RL policy to control or select search operators (Ma et al., 2021; Lu et al., 2019), thereby guiding the search directions towards near-optimal solutions.

**Recent advances in neural VRP.** Recent advancements in neural methods for solving VRPs focus on improving scalability and robustness through innovative architectures and learning strategies. For example, the large-scale performance is improved by employing divide-and-conquer strategies (Fu et al., 2021; Ye et al., 2024), leveraging heavy decoder architectures (Luo et al., 2023), incorporating distance-related bias (Zhou et al., 2024a), and exploiting local transferability (Gao et al., 2024a; Fang et al., 2024); the robustness against distribution shifts is improved by distributional robust optimization (Jiang et al., 2022), multi-distribution knowledge distillation (Bi et al., 2022), meta learning (Zhou et al., 2023) and ensemble learning (Jiang et al., 2023). Furthermore, it is observed that the performance of neural solvers can be enhanced by utilizing a population of complementary models (Grinsztajn et al., 2023; Zhou et al., 2024c; Gao et al., 2024b). Moreover, Liu et al. (2024) proposed to develop a foundation model for a class of VRP variants, leveraging the shared problem structure to achiece better performance. Building on this, Zhou et al. (2024b) further improved model capability by introducing the mixture-of-experts structure. Besides these efforts, this paper focuses on complex constrained VRPs, which are common in real-world applications (Cattaruzza et al., 2017; Glomvik Rakke et al., 2012) but have not received much attention in the research community. Only a few works (Tang et al., 2022; Chen et al., 2024; Bi et al., 2024) try to address it through feature enhancement or Lagrange multiplier method. In this context, we introduce a novel instance-level adpative framework for Lagrangian-based neural methods, reducing the infeasiblity rate significantly.

## C. Instance Generation

In our experiments, we consider two categories of problem, TSPTW and TSPDL. Following prior works (Kool et al., 2019), we randomly sample coordinates $(x_i, y_i)$ for each node $i$ (including the depot) from a uniform distribution $U(0, 1)$ within a square. For generating the time windows and draft limits, we utilize the code of Bi et al. (2024) and adopt the **hard** settings, which are sufficiently challenging to examine state-of-the-art neural and OR solvers. The generation process of time windows and draft limits is detailed as follows.

**Time windows.** After generating the node coordinates, the pairwise travel times are calculated based on the Euclidean distance between any two nodes. For the generation of time windows, we adopt the configuration of a widely recognized benchmark (da Silva & Urrutia, 2010) in our experiments. Specifically, the process begins with the construction of a random tour $\tau$ (i.e., a random permutation of the nodes). Subsequently, the time window $[l_i, u_i]$ for each node $i$ is iteratively generated, where the lower bound $l_i$ and upper bound $u_i$ are uniformly sampled from a range determined by the cumulative travel distance $\phi_i$ of the partial solution up to node $i$ and the maximum window size $2\eta$. More formally, $l_i \sim U[\phi_i - \eta, \phi_i]$ and $u_i \sim U[\phi_i, \phi_i + \eta]$. This procedure guarantees the existence of at least one feasible solution for each instance, and the tight coupling between the time windows and the randomized tours introduces significant complexity to the problem, thereby increasing the computational difficulty of satisfying constraints. In this paper, the maximum window size $\eta$ is set to 50, and we employ a scale factor $\rho = 100$ to normalize the node coordinates and time windows according to (Bi et al., 2024).

**Draft limits.** In the context of TSPDL, each node is associated with a demand value and a maximum draft limit, which is designed to avoid overloaded ships entering these ports (i.e., nodes). From an initial feasible setting, the draft limit of each node is set to the summarized demands of other nodes, thereby ensuring that any node demand can not exceed its own draft limit. Subsequently, a fraction parameter, denoted as $p\%$, is introduced to adjust the draft limits of non-depot nodes. Specifically, $p\%$ of the non-depot nodes are randomly selected, and each of them is assigned a draft limit drawn as a random integer from the range $[\delta_i, \sum_{i=1}^{n} \delta_i]$, where $\delta_i$ is the demand of the $i$-th node. Finally, a feasibility validation is conducted (e.g., utilizing bin-counting constraints) to ensure that the assigned draft limits do not lead to instances without feasible solutions. In our experiment, the node demands are set to 1 and the fraction parameter $p\%$ is set to 90%.

## D. Implementation Details

### D.1. Training Details

The training procedure of our ICO method contains two stages: a pre-training stage and a fine-tuning stage. The pre-training stage involves a total of $10,000$ epochs, while the fine-tuning stage comprises $1,000$ epochs. Each training epoch processes

10, 000 synthetic problem instances. For both stages, we select the model checkpoint that achieves the best inference performance on a validation dataset as the final model. It is worth noting that the training process of our ICO method includes 1, 000 more epochs compared to the training process of POMO+PIP. To ensure a fair comparison, we extend the training of the provided POMO+PIP checkpoints by an additional 1, 000 epochs.

The fine-tuning stage involves the iterative updating of $\lambda$ values. In this process, the initial values $\lambda^{(0)}$ is uniformly set to 0.1 for all problem instances. If the policy fails to find feasible solutions on a specific instance, the $\lambda$ value corresponding to this instance is updated based on the constraint violation, where the learning rate of $\lambda$ is set to 0.5 for TSPTW and 0.2 for TSPDL, since the scales of constraint violations on TSPTW and TSPDL are different. These hyperparameters in updating $\lambda$ are aligned with the corresponding hyperparameters in the inference stage, narrowing the gap of training and inference. To improve computational efficiency and mitigate the risk of overfocusing on challenging instances, the number of iterations is limited to a maximum of 4, and the ratio of infeasible instances within a batch must not exceed 25%. During the fine-tuning on TSPDL50, we observe that the fine-tuned policy tends to overemphasize the constraints, resulting in a near zero infeasibility rate but a significant deterioration in objective values. To mitigate this issue, we adjust the learning rate of fine-tuning process on TSPDL50 to $1 \times 10^{-6}$, while learning rates of other training process remain the default setting (i.e., $1 \times 10^{-4}$).

### D.2. Inference Details

The instance-specific $\lambda$ values are iteratively updated based on constraint violations during the inference stage. In this process, the $\lambda$ values are initialized as 0.1 for all instances except instances of TSPDL100, since it is observed that the conditioned policy fails to obtain feasible solutions for most instances of TSPDL100 when using $\lambda = 0.1$. Consequently, the intial $\lambda$ value for TSPDL100 is increased to 0.5. During the updating process of $\lambda$, the learning rate is configured as 0.5 for TSPTW and 0.2 for TSPDL. These different learning rates are to accommodate the different scales of constraint violations on these two problem types. In the comparison experiments, the number of iterations for updating $\lambda$ is set to 16.

### D.3. Experimental Settings

**Metrics.** Four metrics are applied: Infeasibility rate, average optimality gap, normalized HyperVolumn (HV) and runtime. The instance-level infeasibility rate measures the proportion of instances where the solver fails to find any feasible solution. These metrics are calculated on a test dataset containing 10,000 instances. To compute the optimality gap, we use the solutions obtained by LKH3 through full-time search as reference solutions. Unlike some prior works that compute the optimality gap directly from the average objective (Kool et al., 2019), we calculate the optimality gap on an instance-by-instance basis and then average these values. It is important to note that the calculation of objective values and optimality gaps only includes instances with feasible solutions. Therefore, the average objective value may not serve as a fully reliable metric for performance comparison, as the sets of instances with feasible solutions can vary across different methods. To measure the comprehensive performance of both solution quality and feasibility, we further compute the normalized HV based on the infeasibility rate and average optimality gap. The reference point for computing HV is set to $(100\%, 5\%)$ for TSPTW and $(10\%, 20\%)$ for TSPDL, where the first number represent the infeasibility rate and the other denotes the average gap. To evaluate the computational efficiency, we compare the total runtime of solving 10,000 instances with batch parallelism on a single GPU (NVIDIA RTX 4090 Ti). For OR solvers like LKH3 and OR-Tools, we record the runtime of parallel computation on 16 CPU cores.

**Evalution configurations of baselines.** To align the runtime consumption, POMO+PIP employs $\times 28$ sampling for intances with $n = 50$ and $\times 20$ sampling for instances with $n = 100$, where AM+PIP adopts $\times 200$ sampling for both $n = 50$ and $n = 100$ instances. These different sampling configurations are to align with the additional runtime caused by the computation of $\lambda$-conditioned embeddings in our ICO method. The evaluation batch sizes for both POMO-PIP and our ICO method are set to 2,500 for instances with $n = 50$ and 1000 for instances with $n = 100$.

## E. Additional Results

### E.1. Analysis of Training Strategies

We design a two-stage training strategy that comprises a pre-training stage that makes the policy capable of solving instances with varying degrees of constraint awareness, and a fine-tuning stage that aligns the $\lambda$ values with instance hardness. To

evaluate the effectiveness of these two stages, results of the single-$\lambda$ policy (i.e., POMO+PIP with $\lambda = 1$), the pre-trained policy and final fine-tuned policy are plotted and compared in Figure 5. The results indicate that both the pre-trained policy and the fine-tuned policy achieve superior performance compared to the single-$\lambda$ approach, particularly with respect to the infeasibility rate. Furthermore, the comparison between the pre-trained and fine-tuned policies reveals that the fine-tuning process significantly enhances overall performance, except the slight degeneration in terms of the average gap on TSPDL50. This overall improvement can be attributed to the iterative adjustment of $\lambda$ values, which effectively aligns them with the hardness of specific problem instances.

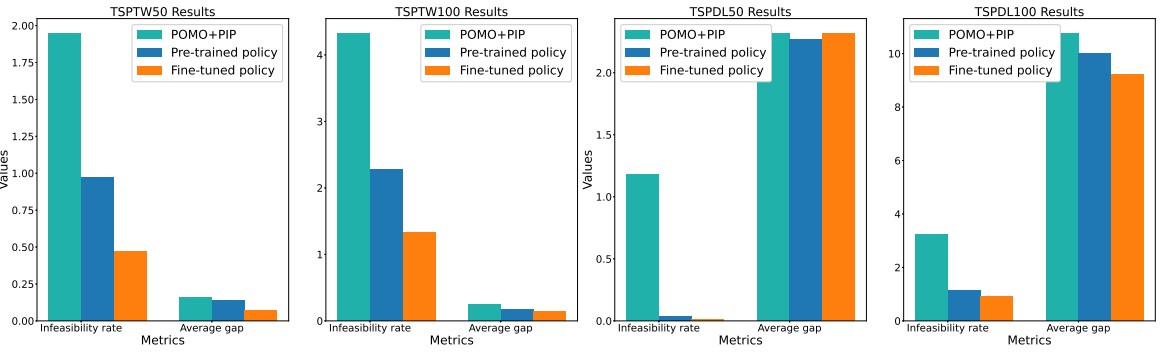

*Figure 5.* Comparison of the pre-trained policy and fine-tuned policy.

### E.2. Analysis of Different Update Rules for $\lambda$

**Proportional-Integral-Derivative (PID) control for updating $\lambda$.** From the perspective of control theory, the subgradient descent process of $\lambda$ behaves as *integral* control, while Stooke et al. (2020) proposed to further incorporate *proportional* and *derivative* control into the update rule, avoiding oscillations encountered by the integral-only controller. The proportional control is to hasten the constraint satisfaction in response to the immediate constraint violation. The derivative control prevents the oscillations by monitoring the variation tendency of constraint violations. By adding the terms of proportional, integral and derivative control, the update rule of PID control is expressed as:

$$
\begin{aligned}
\Delta_t &= g_I(\tau_t), \\
I_t &= I_{t-1} + g_I(\tau_t), \\
\delta_t &= \max\{g_I(\tau_t) - g_I(\tau_{t-1}), 0\}, \\
\lambda_t &= K_P \cdot \Delta_t + K_I \cdot I_t + K_D \cdot \delta_t,
\end{aligned}
$$

where $\Delta_t$ represents the proportional term of time step $t$, $I_t$ denotes the $t$-th step integral term that accumulates the constraint violations of previous steps, $\delta_t$ computes the derivative term of the constraint violation, and $K_P, K_I, K_D$ are tuning parameters that measure the weights of three terms. Intuitively, this PID method provides a richer set of controllers than subgradient descent, but it also introduces more hyperparameters that require manual tuning. In our experiments, $K_P$ is set to 0.1 and $K_D$ is set to 1.0 on both problem types, and $K_I$ is set to 0.5 on TSPTW and 0.01 on TSPDL.

In Table 2, we compare the performance of different update rules of $\lambda$ in inference stage: fixed $\lambda$ values ($\lambda \in \{0.5, 1.0, 2.0\}$), randomly sampled $\lambda$ values, the subgradient descent method and the PID control method (Stooke et al., 2020). For the random sampling strategy, $\lambda$ values are drawn randomly from the uniform distribution $U(0.1, 2.0)$ at each iteration.

The results in the last three rows indicate that both the subgradient descent method and the PID control method generally outperform the random sampling strategy, with particularly improvements in reducing the infeasibility rate. It is worth noting that the random sampling approach also demonstrates competitive performance, indicating that simply varying the $\lambda$ values randomly for each instance can be an effective strategy. Moreover, as evidenced in the first three rows, employing fixed $\lambda$ values leads to significantly inferior performance compared to the adaptive variation of $\lambda$, underscoring the critical importance of dynamically adjusting $\lambda$ for each instance. By comparing the results of the last two rows, it is observed that the PID control method does not achieve superior performance as expected, which can be attributed to two factors: (1) the hyperparameters of PID are challenging to tune; (2) the subgradient descent method is already involved in the fine-tuning process, while the PID control is not integrated into the training, limiting its effectiveness.

*Table 2.* Additional results of different update rules of $\lambda$ on TSPTW and TSPDL. The best results are highlighted in **bold**.

| Methods | TSPTW ($n = 50$) | | TSPTW ($n = 100$) | | TSPDL ($n = 50$) | | TSPDL ($n = 100$) | |
|---|---|---|---|---|---|---|---|---|
| | Inf. rate | Avg. Gap | Inf. rate | Avg. Gap | Inf. rate | Avg. Gap | Inf. rate | Avg. Gap |
| ICO ($\lambda = 0.5$) | 1.43% | 0.19% | 4.34% | 0.26% | 2.63% | 2.50% | 42.14% | 13.16% |
| ICO ($\lambda = 1.0$) | 1.52% | 0.23% | 4.03% | 0.36% | 0.23% | 2.77% | 2.01% | 10.79% |
| ICO ($\lambda = 2.0$) | 1.55% | 0.24% | 4.27% | 0.38% | 0.07% | 3.15% | 0.38% | 11.62% |
| ICO (random) | 0.55% | 0.07% | 2.40% | 0.14% | 0.12% | **2.28%** | 0.40% | 10.73% |
| ICO (subgradient) | **0.51%** | 0.07% | **1.33%** | 0.14% | **0.01%** | 2.32% | 0.91% | **9.22%** |
| ICO (PID control) | 0.55% | 0.07% | 1.39% | 0.14% | 0.05% | 2.36% | **0.26%** | 9.25% |

### E.3. Analysis of Network Architectures

The $\lambda$-conditioned policy network is a key component in our ICO framework, which decouples the policy optimization from the optimization of dual variables. This network should take $\lambda$ as the condition varibable and effectively adjust the constraint awareness according to the varying value of $\lambda$. Among existing network architectures in other domains (Wang et al., 2024; Lin et al., 2022), there are two alternative approaches to implement the conditioned policy: (1) condition $\lambda$ in the initial embeddings; (2) condition $\lambda$ in the decoder's context. The second approach, referred to as the $\lambda$-conditioned context method, is detailed as follows.

$\lambda$**-conditioned context.** Building upon the POMO model (Kwon et al., 2020), the conditioned context method integrates a linear embedding of $\lambda$ into the decoder's *context* embedding, formulated as $\boldsymbol{q} = W^\lambda \lambda + W^q [\boldsymbol{h}^c, t^c]$. Here, $W^\lambda \in \mathbb{R}^{d \times 1}$ and $W^q \in \mathbb{R}^{d \times d}$ are trainable parameters, and $[\boldsymbol{h}^c, t^c]$ denotes the concatenation of the current node embedding $\boldsymbol{h}^c$ and the current time $t^c$, together forming the *context* used for selecting candidate nodes. The resulting output, $\boldsymbol{q}$, functions as the query input for the subsequent multi-head attention layer in the decoder. This conditioned context approach incorporates the information of $\lambda$ into the core component of the decoder, enabling an efficient adjustment of the policy's behavior.

*Table 3.* Additional results of different network architectures on TSPTW and TSPDL. The best results are highlighted in **bold**.

| Methods | TSPTW ($n = 100$) | | TSPDL ($n = 100$) | |
|---|---|---|---|---|
| | Inf. rate | Avg. Gap | Inf. rate | Avg. Gap |
| Network with $\lambda$-conditioned context | 2.83% | 0.30% | 2.31% | 13.34% |
| Network with $\lambda$-conditioned embeddings | **2.28%** | **0.17%** | **1.14%** | **10.01%** |

In Table 3, we compare the performance of the network with $\lambda$-conditioned context and network with $\lambda$-conditioned embeddings on TSPTW100 and TSPDL100. Here we report the results of the pre-trained policies. The experimental results demonstrate that the $\lambda$-conditioned embedding method achieves significantly superior performance in both infeasibility rate and average optimality gap. This performance advantage can be attributed to the fact that the $\lambda$-conditioned embedding utilizes the full capacity of the entire network to process $\lambda$-related information, while the conditioned context approach restricts the processing of $\lambda$-related information to the decoder, thereby limiting its effectiveness.

### E.4. Analysis of Distribution $D(\lambda)$

In the pre-training stage of the conditioned policy, random values of $\lambda$ are sampled from a pre-defined distribution $D(\lambda)$ for training. Empirically, the distribution $D(\lambda)$ has a non-negligible influence on the performance of the pre-trained policy. A natural and straightforward option for $D(\lambda)$ is the uniform distribution within an appropriate range. However, as shown in Figure 6, the trained policy just silghtly violates constraints on the majority of instances, where only a small subset of instances in the long tail experience significant constraint violations. Therefore, we adopt a triangular distribution $T(0.1, 0.5, 2.0)$, which biases the sampling towards smaller $\lambda$ values, thereby prioritizing the optimization of instances with low constraint violations. Figure 7 compares the performance of the policy trained with a uniform distribution $U(0.1, 2.0)$

and the policy trained with a triangular distribution $T(0.1, 0.5, 2.0)$ on the TSPTW50 dataset. The results demonstrate that the triangular distribution leads to superior overall performance as expected.

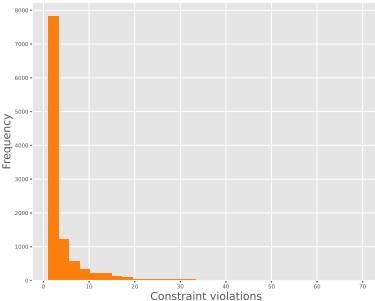

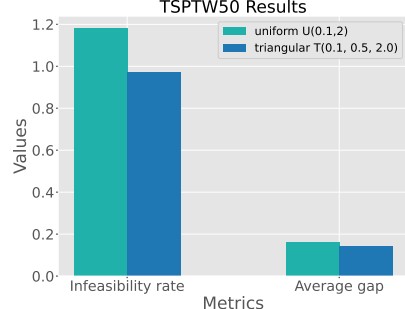

*Figure 6.* Histogram of constraint violation statistics on the validation dataset.

*Figure 7.* Performance of using two different $D(\lambda)$ configurations during the pre-training stage.

### E.5. Analysis of anytime performance

During the inference stage, our ICO method iteratively samples new solutions and updates the $\lambda$ values based on the current constraint violations. Consequently, the anytime performance throughout the iterative process becomes a critical factor. Figure 8 presents the convergence curves of three metrics—hypervolume (HV), infeasibility rate, and average optimality gap—on TSPTW50 and TSPTW100. The results indicate that, while the proposed ICO method exhibits the highest infeasibility rate at the initial stage, it demonstrates a rapid convergence rate and achieves superior performance compared to single-$\lambda$ models in subsequent iterations. Moreover, in terms of HV and optimality gap, the ICO method consistently outperforms single-$\lambda$ models throughout the entire iterative process.

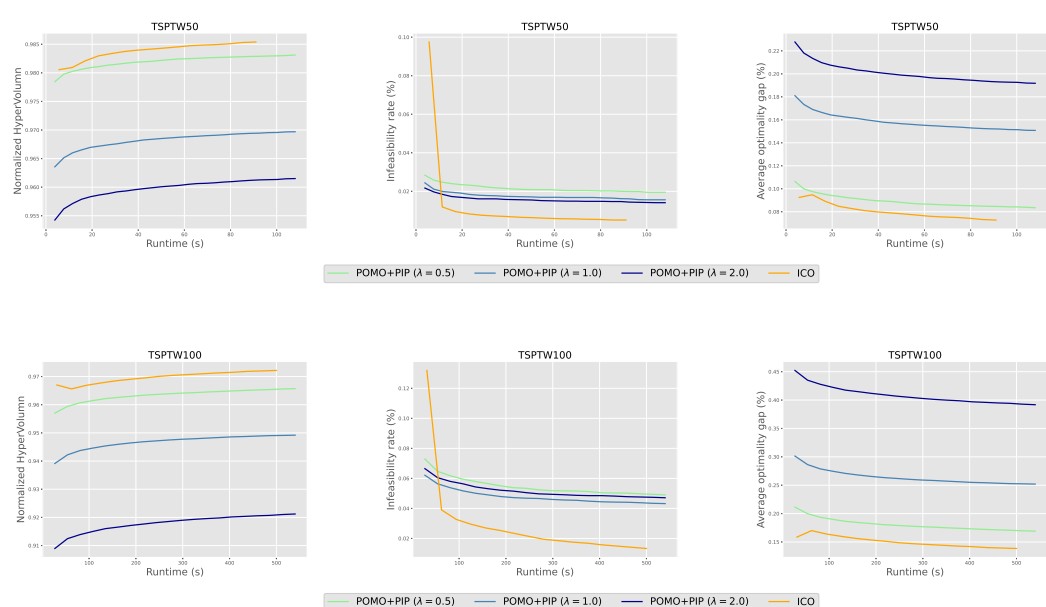

*Figure 8.* Any-performance comparison between our ICO method and the single-$\lambda$ methods. The figures, from top to bottom, represent the convergence curves on TSPTW50 and TSPTW100, respectively.

