# OpenReview forum: "Adaptive Constrained Optimization for Neural Vehicle Routing"
_ICML.cc/2025/Conference — Submitted to ICML 2025_

### Official Review · Reviewer_taQj · 2025-03-09

**Overall Recommendation:** 3

**Summary:**

This work proposes a instance-level adaptive constraint optimization framework to improve the feasibility satisfiability of learning methods for TSPs. The authors designed a dual variable-conditioned policy with two phase of learning. In the first phase, they consider varying values of the lagrange dual variable and lean a model that works for these varying values. in the second phase, they consider an iterative procedure that updates the dual value based on subgradient descent to train the policy on TSP instances that remain infeasible after the first phase training. The authors benchmark their algorithm on TSPTW and TSPDL and observe significant improvement from the prior single-dual value method (PIP).

**Claims And Evidence:**

The claims made in the submission are supported by clear and convincing evidence. The authors show that their method significantly improve from the previous work (PIP) in terms of infeasibility rate.

**Essential References Not Discussed:**

N/A

**Experimental Designs Or Analyses:**

The experimental designs in this paper is sound and valid. The authors provide detailed ablation studies (e.g. contribution of different training stages, comparison of different update rules, and analysis of network architecture) to justify their design.

**Methods And Evaluation Criteria:**

The proposed methods and evaluation criteria make sense for the problem.

**Other Comments Or Suggestions:**

N/A

**Other Strengths And Weaknesses:**

Other Strength:

1. I find the paper to be well written. I especially enjoyed reading the introduction section. I find Figure 1 very clear and motivates the proposed method very well.
2. The authors demonstrate that the proposed learning method has good improvement from PIP.

Other Weakness

1. The authors discuss vehicle routing many places in the paper (e.g. title, abstract, section 2.1). However, their experiments are only on TSP variants (not VRP). It feels a little weird to claim that this works solve vehicle routing problems.
2. There's no theoretical analysis of the paper.
3. The idea of adapting the dual variable to improve feasibility is not entirely novel, and Tang 2022 seems to have a similar idea proposed. Can the authors comment on the difference with Tang 2022?

Tang, Qiaoyue, et al. "Learning to Solve Soft-Constrained Vehicle Routing Problems with Lagrangian Relaxation." arXiv preprint arXiv:2207.09860 (2022).

**Questions For Authors:**

1. In Sec 2.1 the draft limit constraint in TSPDL looks similar to the time window constraints in TSPTW. Can the authors comment on the differences of these two constraints, e.g. will they lead to different solving difficulty? Furthermore, given that for most of the TSPTW/TSPDL instances tested here, the infeasibility rate for POMO + PIP is already very low, I wonder if the authors can include results of exploring other TSP / VRP variants with higher infeasibility rate to further demonstrate the performance improvement of their method?
2. I wonder how the dual variable-conditioned policy performs without applying PIP masking from the previous literature?
3. Phase 1 solve the inner subproblem: the authors train a single policy for different values of lambda. Can the authors comment on whether they observe that learning is effective for a certain subset of lambda values but not for others?
4. Figure 3 (middle): the infeasible part / infeasible values labels for the purple cell: I'm a bit confused what they mean. Do the authors mean those are the Lagrangian values lambda that make the solution infeasible?
5. For the fine-tuning and inference stage, when the authors update the parameters lambda, I'm curious if the authors see any patterns in the lambda update for different instances? That is, do the authors observe a common sequence of updated lambdas that work well for a lot of instances, or the lambda values tend to be different after a few subgradient updates?

**Relation To Broader Scientific Literature:**

This work advances the scientific community by proposing an adaptive dual variable-conditioned policy, whereas previous literature only uses a single dual variable. The proposed method can significantly improve the performance for solving hard TSP instances, which is important for the optimization community.

**Theoretical Claims:**

The paper does not provide any theoretical claims or proofs.

---

> ### Author Rebuttal · Authors · 2025-04-01
>
> Thanks for dedicating your time to review our work! We sincerely appreciate your acknowledgment of our contributions. Below are the detailed responses to your concerns. **Corresponding experimental results** can be found at [link](https://anonymous.4open.science/api/repo/4620_rebuttal-DDDF/file/addtional_results_4620.pdf).
>
> ## **Response 1: Addressing the relationship between TSP variants and VRP (for weakness 1) and extending to more problem variants (for question 1)**
>
> Thank you for your thoughtful feedback. Recent studies in neural combinatorial optimization (Kwon et al., 2020; Bi et al., 2024) often view TSP as a single-vehicle variant of VRP, capturing core route optimization challenges. Studying TSP variants also offers insights into solving multi-vehicle VRP, which can be decomposed into vehicle assignment and route planning, where each vehicle solves a TSP independently.
> Thanks to your suggestion, we have extended our experiments to multi-vehicle VRP variants, such as CVRPTW. Detailed analyses are provided in **Response 5 to Reviewer X7da** due to character limits.
>
> ## **Response 2:  Difference with Tang et al, 2022 (for weakness 3)**
>
> Thank you for your insightful comments. As shown in Equation (2) of Tang et al. (2022), their method ties dual variables to the policy, forming a policy-level $\lambda$ approach. This is conceptually equivalent to the single-$\lambda$ method, where $\lambda$ updates with policy changes but remains invariant across instances. In contrast, our proposed instance-level $\lambda$ method introduces a key distinction: it decouples the dual variable from policy optimization, allowing $\lambda$ to adapt dynamically to each instance and better handle instance-specific variations.
>
> ## **Response 3: Difference between time window and draft limit constraints (for question 1)**
>
> Thank you for your valuable question. While these constraints share some similarities, their properties differ. The time window constraint is theoretically more challenging (NP-complete) and empirically shows higher infeasibility rates than the draft limit constraint. TSPDL, however, also has intriguing properties. Although its constraints can be easily satisfied using a Greedy-C heuristic, balancing its objective value and constraint violations remains highly challenging. For example, Table 1 shows that increasing $\lambda$ in TSPDL50 significantly impacts the infeasibility rate and optimality gap. By contrast, the performance variation on TSPTW50 under similar conditions is relatively minor. This observation highlights TSPDL's heightened sensitivity to a method's ability to balance trade-offs, making it a valuable and distinctive benchmark. Thus, both TSPTW and TSPDL are essential for our experiments, offering complementary insights for evaluation.
>
> ## **Response 4: Experimental results without PIP mask (for question 2)**
>
> Thank you for your insightful question. We have conducted additional TSPTW50 experiments without the PIP mask, as shown in **Table S6**. Removing it significantly increases problem difficulty, creating a more challenging benchmark. Interestingly, the results reveal that our ICO method achieves a greater performance improvement under this more challenging setting, further highlighting the superiority of the proposed instance-level adaptive approach.
>
> ## **Response 5: Whether the learning in phase 1 is effective on a certain subset of $\lambda$ (for question 3)**
>
> Good point! Yes, the learning is only effective when the range of $\lambda$ is reasonable, i.e., on a certain subset of $\lambda$. In the experiments on TSPDL, we observed that when $\lambda$ exceeds 10 or falls below 0.1, the policy often fails to converge. But fortunately, the reasonable range ($0.1 \le \lambda \le 2.0$) is simple to identify through heuristic trials. If you are curious about more $\lambda$ settings, please see **Response 2 to Reviewer LzKt**.
>
> ## **Response 6: Description of Figure 3. (for question 4)**
>
> The “infeasible part” refers to infeasible instances and their $\lambda$ values (purple in Figure 3), while “infeasible values” denote their constraint violations. We clarify “retain the infeasible part” in lines 250–253 and will revise the paper to avoid misunderstandings.
>
> ## **Response 7: Patterns of $\lambda$ update (for question 5)**
>
> Thank you for your thoughtful question. Yes, we observe some common patterns in $\lambda$ updates for certain instances. We refer you to Figure 6 for the information that many infeasible solutions exhibit only minor constraint violations. For this subset of slightly infeasible instances, their $\lambda$s will quickly converge to some relatively small values within 2-3 iterations and remain stable in the subsequent iterations, which is a common pattern, but their updated $\lambda$ values are not exactly the same. In contrast, for instances with larger violations, no common pattern emerges, and $\lambda$ values diverge based on violation magnitude.

---

### Official Review · Reviewer_iCcJ · 2025-03-13

**Overall Recommendation:** 3

**Summary:**

This paper incorporates an instance-adaptative Lagrangian multiplier into the policy of the neural VRP solvers, aiming to enhance the performance of a most recent work PIP (Bi et al., 2024), which uses a fixed multiplier during training. Specifically, the multiplier varies among different instances and inputs as the feature for each instance. During training, it first trains the PIP model with randomly sampled multipliers, and then finetunes the model using infeasible instances with updated multipliers. During inference, the multipliers for each instance are updated via a subgradient method to generate solutions. The proposed method is tested on TSPDL and TSPTW with 50 and 100 nodes.

---
## First round rebuttal:
Concerns regarding computational overhead, insufficient experiment and limited novelty -> maintain the score

## Second round rebuttal:
Thanks for the response. Most of my concerns has been resolved. I feel like the novelty is moderate, so if other reviewers lean toward acceptance, I am also okay with accepting this paper. I strongly encourage the authors to include all additional results in the revised manuscript, particularly the discussion on computational overhead. Moreover, the advantage of encoding the multiplier into a hypernetwork should be clearly differentiated and clarified in comparison to existing primal-dual methods, such as the following:

[1] Qiaoyue Tang, Yangzhe Kong, Lemeng Pan, and Choonmeng Lee. Learning to solve soft-constrained vehicle routing problems with lagrangian relaxation. arXiv preprint, 2022.
[2] Stooke, Adam, Joshua Achiam, and Pieter Abbeel. Responsive safety in reinforcement learning by pid lagrangian methods. ICML, 2020.
[3] Weiqin Chen, et al. Adaptive Primal-Dual Method for Safe Reinforcement Learning. AAMAS, 2024.
[4] Park, Seonho, and Pascal Van Hentenryck. Self-supervised primal-dual learning for constrained optimization. AAAI, 2023.

**Claims And Evidence:**

Yes, mostly.

**Essential References Not Discussed:**

There are many recent works studying effective updates of the Lagrange Multiplier Method. However, this paper lacks a literature review on this aspect.

**Experimental Designs Or Analyses:**

Yes. See weaknesses and questions.

**Methods And Evaluation Criteria:**

Yes.

**Other Comments Or Suggestions:**

Typo: LKH (less time) should be coloured in blue on TSPDL100.

**Other Strengths And Weaknesses:**

**Strengths:**
1. This paper enhances the constraint-handling capability of the neural VRP solver, which is a significant challenge.
2. The proposed method obtains better performance compared to the state-of-the-art method PIP.
3. The code is provided.

**Weaknesses:**
1. The proposed method incurs significantly higher computational overhead than PIP due to the need to update the multipliers during both training and inference.
2. The experiments are insufficient, and some implementation details are missing.
3. The method's applicability is limited to TSPTW and TSPDL.
4. The writing quality requires improvement in terms of accuracy, consistency with the source code, and clarity, particularly regarding notations and font/table sizes.

**Questions For Authors:**

1. What is the training time of the proposed method? Why is the inference time for instances with $n=100$ so prolonged? Instead of extending the inference time of PIP through sampling, have you considered alternative post-search strategies such as SGBS [1] and EAS [2]? The current comparison with PIP may be unfair.
2. Can the proposed method be extended to handle broader TSP/VRP variants beyond the current scope?
3. Why do the randomly sampled multipliers work in the pre-train stage (for the first 10000 epochs)? I understand the motivation of applying different multipliers for each instance due to different constraint violations, but the randomly sampled ones do not reflect this kind of violation information. Why does it work? Any deeper explanations or theoretical justifications?
4. Does the effectiveness and efficiency of the fine-tuning stage depend on the initial value of the multiplier? Why was 0.1 chosen as the initial value? What would happen if other values were used?
5. It seems in the source code TSPDL uses the multipliers randomly sampled from a Gaussian distribution during the pre-training stage, which is inconsistent with description in the main paper. Why using it on TSPDL? What is the performance variance when using different distributions beyond the results presented in Appendix E.4?
6. Why does OR-Tools achieve inference times of only a few seconds?
7. What is the performance variance when different learning rates are applied during inference?
8. Does Phase 2 only use the constraint violation $g$, while ignoring the penalty of the timeout nodes $c$ used in Phase 1? Why? Please also check the inconsistency between the equations in lines 256-261 and line 686.

[1] Simulation-guided Beam Search for Neural Combinatorial Optimization. NeurIPS'22.

[2] Efficient Active Search for Combinatorial Optimization Problems. ICLR'22.

**Relation To Broader Scientific Literature:**

This paper is an improvement on the recent work PIP (Bi et al., 2024), aiming to enhance the constraint-handling capability for neural VRP solvers. Specifically, this paper extends PIP from a single Lagrangian multiplier to an instance-adaptive one.

**Theoretical Claims:**

N/A

---

> ### Author Rebuttal · Authors · 2025-04-01
>
> Thank you for your valuable comments and recognizing our contributions. Below please find our responses. **Corresponding experimental results** can be found at [link](https://anonymous.4open.science/api/repo/4620_rebuttal-DDDF/file/addtional_results_4620.pdf).
>
> ## **R1: Experiments with post-search strategies (for Q1 and W2)**
>
> Thanks for your insightful questions! The post-search strategies such as EAS can integrate well with our proposed framework. Thanks to your suggestion, we have revised to extend the TSPTW experiments with EAS beyond basic sampling. As shown in **Table S4**, EAS can enhance both PIP and ICO. Our ICO + EAS still holds superior performance compared to the best results of PIP + EAS. This further validates the superiority of our proposed instance-level adaptive method. We will include these results into our revised paper. Thank you very much for your valuable comments.
>
> ## **R2: Runtime of ICO (for Q1 and W1)**
> Please refer to **R3 to Reviewer LzKt** due to space limitation.
>
> ## **R3: More problem variants (for Q2 and W3)**
> Please refer to **R5 to Reviewer X7da** due to space limitation.
>
> ## **R4: Reasons for using randomly sampled $\lambda$ in pre-training (for Q3)**
> The rationale of using random $\lambda$ can be explained from two perspectives:
> 1. Training Efficiency: In the early training stage, the policy is unconverged and generates many infeasible solutions, making instance-specific $\lambda$ updates computationally expensive. Random sampling is a practical alternative. Once the policy has sufficiently converged, the computational cost of updating $\lambda$ for the few remaining infeasible instances becomes more manageable. At this point, the training can transfer to the next stage, where instance-specific $\lambda$ values are iteratively optimized.
> 2. Enhanced Generalization for Inference: The $\lambda$-conditioned policy aims to behave optimally under any given $\lambda$ values. By exposing the policy to a wide range of random $\lambda$ values, its generalization ability to unseen $\lambda$ is improved. This enhanced generalization is critical for robust performance in the inference stage, where the policy cannot be retrained for every $\lambda$ value.
>
> ## **R5: Sensitivity of $\lambda$ (for Q4 and Q7)**
> Please refer to **R2 to Reviewer LzKt** for sensitivity analysis during inference. The hyperparameters used during fine-tuning are configured to align with inference.
>
> ## **R6: Inconsistency of the Gaussian Distribution of $\lambda$ (for Q5 and W4)**
> When collecting the running scripts for submission, we accidentally pasted the shell file from our earlier attempts into the final train_ICO.sh, where a Gaussian distribution of $\lambda$ appears in the pre-training configuration. We sincerely apologize for this careless mistake and the subsequent misunderstanding. We confirm that the settings of our experiments **are entirely consistent** with our description in the README file and lines 325-327 in the paper, i.e., the results on both TSPTW and TSPDL are trained using the triangular distribution.
>
> In our early experiments, a Gaussian distribution $N(0.1,1.0)$ and a triangular distribution $T(0.1, 0.5, 2.0)$ were tested to emphasize small $\lambda$ values. We provide the results on TSPTW50 in **Table S5**. The simple T(0.1, 0.5, 2.0) has the best performance, which is chosen as the default setting for the final experiments. Besides, all the tested distributions outperform the PIP baseline, further verifying the robustness of our method.
> We will include these discussions in our revised paper. Thank you very much.
>
>
> ## **R7: ORTools (for Q6)**
>
> Thank you for your thoughtful question. The algorithm of OR-Tools generally consists of two steps: (1) generating feasible solutions via greedy heuristics and (2) iteratively refining them. However, for TSPDL, the default greedy heuristic fails to produce feasible solutions. Therefore, the optimization terminates in just a few seconds, without any feasible results.
>
> ## **R8: Ignored $c$ (for Q8)**
>
> The heuristic reward $c$ is also utilized in Phase 2. In Section 3.1, where we provide an overview of the proposed framework, the function $g$ is introduced as a conceptual definition that accounts for all constraint violations, and its abstract nature allows the inclusion of $c$. Explicitly adding $c$ to the equations in Section 3.1, however, may compromise the simplicity and readability of the overview. We appreciate your feedback and will revise the paper to address the inaccuracies in the Appendix.
>
> ## **R9: In Essential References: "recent works studying effective updates of the Lagrange Multiplier Method"**
>
> Thanks for your comment. We have discussed and compared the effective updates method such as subgradient and PID-based update in Appendix E.2. If there are any omissions, please let us know; we are happy to discuss and compare them.
>
> ---
>
> **We hope these can address your concerns, but if we missed anything please let us know.**

---

> > ### Comment · Reviewer_iCcJ · 2025-04-07
> >
> > Thank you for the rebuttal. However, I still believe that the computational overhead of ICO is significant, as it requires iterative updates of the multipliers during inference (with 16 iterations). As PIP is already computationally heavy, the proposed ICO further amplifies this disadvantage. Specifically, PIP requires 15s and 48s for TSPTW50 and TSPTW100, respectively, whereas ICO takes 91s and 8 minutes. I am also not convinced by the comparison to PIP that involves simply adding sampling. What if a lightweight LKH were added to PIP instead, compared to the prolonged ICO? More importantly, I believe the contribution of incorporating different Lagrangian multipliers into PIP to be relatively minor. To this end, I maintain my current score.

---

> > > ### Author Response · Authors · 2025-04-09
> > >
> > > Thank you for taking the time to review our paper. We fully understand your concerns regarding inference overhead. However, we kindly refer you to the **anytime performance analysis (see Appendix E.5)** and the **experiments with EAS (see Table S4)**, which were already provided and emphasized in our initial response. We believe these results could address most of your remaining concerns. In light of your new comments, we would like to further clarify the inference overhead issue and reiterate the key contributions of our work as follows:
> > >
> > > ## Response to “PIP requires 15s and 48s for TSPTW50 and TSPTW100, ..., ICO takes 91s and 8 minutes. ... not convinced ... PIP that involves simply adding sampling.”
> > >
> > > We would like to emphasize once again that, as shown in Appendix E.5, **our proposed method still has superior performance with significantly less runtime than “91s and 8 minutes”**—a point we had already underscored in our previous response. In light of your new comments, we provided experimental results with less runtime in Tables R1 and R2 at [new_link](https://anonymous.4open.science/api/repo/4620_rebuttal-DDDF/file/new_results.pdf). The results demonstrate that our method requires **only 2 iterations (samples)**, which **consumes only 14s on TSPTW50 and 73s on TSPTW100**, to outperform PIP with 20 (28) samples, further verifying the efficiency and effectiveness of our approach.
> > >
> > > Regarding your concern that "PIP with simple sampling" may not be a convincing baseline, we fully agree. However, **we have already conducted new experiments with advanced post-search strategies (i.e., EAS) to address your similar concerns in our prior response**. The results in Table S4 demonstrate that **ICO + EAS still consistently outperforms PIP + EAS**.
> > >
> > > ## Response to “What if a lightweight LKH were added to PIP instead, compared to the prolonged ICO?”
> > >
> > > Good point! In response, we conducted new experiments comparing PIP (greedy) + LKH3 with our ICO (sampling) method, shown in Tables R1 and R2 at [new_link](https://anonymous.4open.science/api/repo/4620_rebuttal-DDDF/file/new_results.pdf). The results on TSPTW50 and TSPTW100 (see the tables below) show that ICO (sampling) significantly outperforms PIP (greedy) + LKH3 in terms of infeasibility rate, while PIP (greedy) + LKH3 achieves the best optimality gap.
> > >
> > > These results indicate that even adding a strong post-search such as LKH3 to the baseline, **our ICO method remains superior in reducing infeasibility**, which is critical for constrained optimization tasks. Moreover, it is important to note that **our ICO (sampling 2) can also be combined with LKH3**.
> > >
> > > If you have more concerns about the high infeasibility rate of **ICO (greedy)**, we want to emphasize that we use a very small initial value of 0.1. When this value is increased, the infeasibility rate can be significantly reduced. Furthermore, we will explore training a hard instance classifier in future work to adaptively decide the initial value for each instance, which would significantly improve the one-iteration performance.
> > >
> > > ## Response to “More importantly, I believe the contribution of incorporating different Lagrangian multipliers into PIP to be relatively minor.”
> > >
> > > We respectfully disagree. Existing methods (not just PIP) tie the Lagrangian multiplier to the policy parameters, **not to the optimization problem itself**. This leads to a fundamental issue, i.e., overlooking the fact that constraint violations vary across problem instances.
> > >
> > > In a strict sense, such methods **cannot be considered proper Lagrangian methods**, as they misalign the dual variables. Our work addresses this long-standing issue by introducing **instance-level adaptive dual variables**, which we believe is a conceptually significant advancement for the NCO community.
> > >
> > > Thus, our contribution should not be viewed as a mere incremental improvement over PIP, but as **an orthogonal and independent innovation** that addresses a critical shortcoming of existing approaches.
> > >
> > > ## Response to “PIP is already computationally heavy, ...”
> > >
> > > We acknowledge that PIP itself is computationally expensive—the complexity of the one-step PIP mask is $O(N^3)$, where $N$ is the number of nodes. However, it is important to clarify that our proposed method of **instance-level adaptive dual variables is general, not tied to the PIP mask**, and therefore **not inherently computationally heavy**.
> > >
> > > For example, we conducted experiments **without using the PIP mask** in response to Reviewer taQj (please see Table S6). The results demonstrate that our method works effectively on top of a light-weight baseline like POMO, achieving a **15.36% reduction in infeasibility rate**.
> > >
> > > Moreover, as discussed with Reviewer X7da, our approach has the potential to be extended beyond neural combinatorial optimization, for example to domains like safe reinforcement learning.
> > >
> > > ---
> > > Thank you for your comments. We appreciate the opportunity to elaborate on these important points.

---

### Official Review · Reviewer_X7da · 2025-03-14

**Overall Recommendation:** 3

**Summary:**

This paper extends the PIP framework (Bi et al., 2024) by allowing the assignment of distinct dual variables to accommodate varying constraint satisfaction difficulties across instances. To achieve this, the paper first proposes modifying the POMO network to incorporate dual variable information into the node embeddings. It then introduces a two-stage training framework to effectively learn the policy. Experimental results on two constrained VRP variants demonstrate that the proposed method improves feasibility handling performance.

**Claims And Evidence:**

Most of the claims are supported well by evidence.

**Essential References Not Discussed:**

None.

**Experimental Designs Or Analyses:**

One major concern regarding the experiments is that the current study focuses solely on improving the PIP method.

It would be more comprehensive to explore whether the proposed ICO framework can enhance other Lagrangian-based methods in neural combinatorial optimization (NCO), potentially extending beyond VRP. Also, it would be interesting to investigate whether ICO can be applied to more general constraint-handling tasks, such as those in safe reinforcement learning?

Another drawback is the time complexity. It is recommended to report and compare the training time, as the proposed training approach may introduce additional computational overhead. How can the time complexity in both training and inference be mitigated?

In Table 2, ICO (random) already exhibits strong performance. Providing an explanation and discussion regarding this observation would be beneficial.

**Methods And Evaluation Criteria:**

The two-stage training approach and the strategy for decoupling policy optimization from dual variable optimization appear to be novel and effective. Also, the dual variable search during inference enhances the flexibility of handling hard constraints. The experimental results demonstrate reasonable improvements over the most recent approaches for constraint handling in TSPWT and TSPDL.

Regarding the update rule for the dual variable (presented just before Section 3.2), I would like the authors to elaborate on its connection to the objectives in Equation (1). Can this update rule be directly derived from Equation (1)?

**Other Comments Or Suggestions:**

Regarding the "update $\lambda$ and retain the infeasible part" process shown in Figure 3, its explanation in the main paper is unclear. It would be helpful to clarify its meaning and explicitly describe how it relates to the purple color in Figure 3.

Additionally, many important results are placed in the appendix rather than the main text. The paper could be reorganized to include key results in the main body for better readability.

**Other Strengths And Weaknesses:**

The paper presents and explores a promising approach to enhancing constraint handling for VRPs. However, the main limitation is its narrow focus on improving a specific method, PIP. Including discussions or experiments demonstrating the generalizability of the proposed method beyond PIP and even beyond NCO would be a valuable addition.

**Questions For Authors:**

Please refer to the above.

**Relation To Broader Scientific Literature:**

The paper may benefit constraint handling in NCO and other domains that require Lagrangian methods and learning-based approaches.

**Theoretical Claims:**

Not applicable.

---

> ### Author Rebuttal · Authors · 2025-04-01
>
> Thanks for dedicating your time to review our work! We sincerely appreciate your recognition of our contributions to constrained combinatorial optimization. Detailed responses to your concerns are as follows. **Corresponding experimental results** can be found at [link](https://anonymous.4open.science/api/repo/4620_rebuttal-DDDF/file/addtional_results_4620.pdf).
>
> ## **Response 1: The derivation of the $\lambda$ update rule**
>
> Thank you for your thoughtful question. The update rule for $\lambda$ presented in Section 3.2 is indeed derived from the objective in Equation (1), albeit in a simplified form. Specifically, the subgradient of Equation (1) with respect to the dual variable $\lambda_{I_i}$ can be obtained by directly computing the derivative. This yields the expression This yields the expression $\mathcal{J}\_C(\pi_{\theta^*}, I_i) = 𝔼\_{\tau \sim \pi_{\theta^*}(\cdot | I_i)}[-g\_{I_i}(\tau)]$, where $\theta^*$ denotes the optimal policy parameters corresponding to $\lambda_{I_i}$. Based on it, our update rule has two simplifications: (1) Our policy is optimized but not guaranteed optimal; (2) The expectation in the subgradient should ideally be estimated using the Monte Carlo method, but in our inference setting, the policy only samples one solution at each iteration for computational efficiency, i.e., we simplify the average estimation to a single value, $g_I(\tau_{t-1})$. We will revise the paper for clarity.
>
> ## **Response 2: Time complexity**
>
> Thanks for your valuable questions! Due to character limitations, please refer to **Response 3 to Reviewer LzKt** for a detailed explanation. In conclusion, the training time of our method is approximately the same as the PIP baseline, and our method can outperform PIP even with less inference runtime.
>
> ## **Response 3: Why ICO (random) performs well**
>
> Thank you for your insightful question! Unlike constrained RL, ICO (random) does not fine-tune the policy's parameters, so there is no need to enforce smooth $\lambda$ updating to mitigate fluctuations, making randomly sampled $\lambda$ values acceptable. The strong performance may stem from the conditioned policy's generalization ability. During pre-training, the policy is exposed to a wide range of randomly sampled $\lambda$ values, allowing it to generalize well across diverse $\lambda$ settings and adapt seamlessly to the random update rule during inference. Additionally, the random update rule itself is not inherently weak. By evaluating all sampled $\lambda$ values and selecting the best-so-far solution, it has a reasonable ability of identifying an effective $\lambda$.
>
> ## **Response 4: Clarification on Figure 3**
>
> Thank you for your insightful comments! The term “infeasible part” refers to the infeasible instances and their associated $\lambda$ values, which are depicted in purple in Figure 3. The phrase “retain the infeasible part” is explained in detail in lines 250–253 of the paper. We apologize for any confusion caused by the use of the term “infeasible part” and will revise the paper to ensure clarity and avoid potential misunderstandings.
>
> ## **Response 5: Extension to more problems/tasks**
>
> Thank you for your valuable comments and questions! The idea of instance-level adaptive dual variables is not specially designed for TSPTW and TSPDL; rather, it can be extended to other domains that simultaneously require constraint handling and cross-instance (or cross-environment) generalization of the RL policy, with domain-specific adaptations. To demonstrate generality, we have revised to add more VRP variants to our experiments. Below is a summary of hard-constrained problems addressed in prior works:
>
> | Related work | TSPTW | TSPDL | CVRPTW |
> | --- | --- | --- | --- |
> | MUSLA (Chen et al., 2024) | Yes | No | No |
> | JAMPR (Falkner & Schmidt-Thieme, 2020) | No | No | Yes |
> | Chen et al., 2022 | No | No | Yes |
> | Tang et al., 2022 | Yes | No | Yes |
> | PIP (Bi et al., 2024) | Yes | Yes | No |
> | Ours | Yes | Yes | Yes (in rebuttal) |
>
> CVRPTW is the only problem not addressed in our experiments. While the decision space of CVRPTW appears more complex, it is, in fact, easier to satisfy its constraints compared to TSPTW and TSPDL. This is because its time window constraints can be easily satisfied by a shortcut: Add more vehicles. To construct a challenging benchmark, we propose to set a maximum limit on the number of vehicles, which also aligns more closely with real-world applications.
>
> We have revised to conduct new experiments on CVRPTW50 with limited vehicles using JAMPR's time window generation code. Since PIP has not been extended to this problem, we used POMO as the backbone to implement ICO. Experimental results in **Table S3** show that our ICO significantly outperforms the POMO baseline, especially in infeasibility rate. Full results will be included in the final paper. Thank you once again for your insightful questions. We hope these new results address your concerns.

---

> > ### Comment · Reviewer_X7da · 2025-04-05
> >
> > Thank you for your rebuttal! I am keeping my positive score.

---

> > > ### Author Response · Authors · 2025-04-09
> > >
> > > Thank you very much for your positive feedback. We sincerely appreciate your insightful comments and constructive suggestions, such as runtime issues and the inclusion of more tasks, which are valuable in guiding the improvement of our work. We will carefully revise the paper to incorporate these important aspects.
> > >
> > > We also appreciate your observation that our proposed approach may have potential applications beyond combinatorial optimization, such as in safe reinforcement learning. We fully agree with this perspective and are willing to explore this promising direction in our future research.

---

### Official Review · Reviewer_LzKt · 2025-03-19

**Overall Recommendation:** 3

**Summary:**

The paper introduces an instance-level adaptive constrained optimization method for neural vehicle routing. It builds on prior PIP-based approaches by assigning instance-specific dual variables instead of a uniform λ, aiming to better balance solution quality and constraint satisfaction across diverse instances. The proposed two-stage training—comprising a pre-training phase with randomly sampled λ values and a fine-tuning phase with iterative dual updates—shows promising improvements in infeasibility rates and solution quality on TSPTW and TSPDL benchmarks.

## update after rebuttal

The authors have not fully addressed the concerns I raised in my initial review. Consequently, I have decided to maintain my original score.

**Claims And Evidence:**

The claims are generally supported by experimental evidence. The authors show significant reductions in infeasibility rates compared to both state-of-the-art neural methods and strong OR solvers. However, the reliance on iterative updates based on PIP appears to introduce considerable runtime overhead, which is a serious drawback that might affect practical deployment. Moreover, while the overall improvements are clear, the paper falls short in dissecting the individual contributions of each component. In particular, the sensitivity of the method to different λ values is not thoroughly analyzed in the ablation studies, leaving some uncertainty about the robustness of the proposed mechanism.

**Essential References Not Discussed:**

N/A

**Experimental Designs Or Analyses:**

Experimental designs are well laid out with comparisons against both neural and traditional OR solvers. The results show marked improvements in feasibility and competitive optimality gaps. Yet, the experimental section does not fully explore how sensitive the outcomes are to the choice of λ or how each part of the model contributes to the final performance. A more comprehensive ablation study is needed to confirm that the improvements are not solely due to the increased iteration time rather than genuine methodological advances.

**Methods And Evaluation Criteria:**

The methods and evaluation criteria are appropriate for the problem. The idea of decoupling the dual variable optimization from policy optimization is sound and innovative. Yet, the increased computational cost due to multiple iterations for updating instance-specific λ values is concerning. The paper would benefit from a more detailed analysis of how this trade-off impacts overall performance, especially in time-sensitive applications.

**Other Comments Or Suggestions:**

N/A

**Other Strengths And Weaknesses:**

N/A

**Questions For Authors:**

I suggest the authors clarify the trade-off between performance gains and computational overhead. Could they provide more detailed ablation studies to isolate the impact of each component, particularly the sensitivity of the method to different λ settings? Additionally, how do they justify the iterative updates given the significant increase in runtime? Answers to these questions might change the overall evaluation of the method's practicality.

What are your thoughts on addressing the runtime overhead issue without sacrificing the performance gains? Also, could further experiments help clarify the exact contributions of each module in your framework?

**Relation To Broader Scientific Literature:**

The work is well-situated within the broader literature on neural VRP and constrained optimization. It builds on recent advances and provides meaningful comparisons with existing state-of-the-art methods. Nonetheless, some recent studies on adaptive dual variable strategies and dynamic constraint management in combinatorial optimization are not cited, which could help contextualize the contributions more robustly.

**Theoretical Claims:**

Theoretical claims in the work are mainly heuristic. Although the paper explains the motivation behind the dual conditioning and provides a clear training algorithm, there is limited discussion on the theoretical convergence properties of the iterative process or on the stability of the dual updates. This gap might affect the confidence in the method's long-term performance.

---

> ### Author Rebuttal · Authors · 2025-04-01
>
> Thank you for reviewing our work and recognizing our contributions. Below are detailed responses. **Corresponding experimental results** can be found at [link](https://anonymous.4open.science/api/repo/4620_rebuttal-DDDF/file/addtional_results_4620.pdf).
>
> ## **Response 1: Lacking ablation study**
>
> Due to space constraints of the main paper, most of our ablation results were included in Appendix E, including the two-stage training strategy, the updating rule of $\lambda$, the conditioned network architecture, and the prior distribution of $\lambda$. Section 4.3 briefly summarizes these findings, showing the effectiveness of each component. To improve clarity, we will integrate key results into the main text.
>
> ## **Response 2: Sensitivity of $\lambda$**
>
> Thank you for your valuable feedback. Since the optimization landscape for $\lambda$ is typically non-convex due to the hardness of combinatorial optimization, we agree that both the initial value and learning rate of $\lambda$ are important for the optimization performance. Thanks to your suggestion, we have revised to conduct a sensitivity analysis of $\lambda$ from these two perspectives, including the initial value of $\lambda$ (denoted as $\lambda_0$) and the learning rate for updating $\lambda$ (denoted as $\alpha$). During the inference stage, we evaluated performance across $\lambda_0 \in \\{0.1, 0.15, 0.20, 0.5, 1.0\\}$ and $\alpha \in \\{0.1, 0.2, 0.5, 0.7, 1.0\\}$ on TSPTW50 and TSPDL50. Each hyperparameter was varied while keeping the other fixed at its default value. Results in **Table S1 and S2** show that:
>
> 1. In 16 out of 18 settings, our ICO method surpasses the best-performing PIP model in hypervolume (HV), showing its robustness.
>
> 2. Although the performance variance (shown in the last row) is relatively small, it is not negligible. This underscores the importance of carefully tuning $\lambda$-related hyperparameters to achieve optimal performance.
>
> Interestingly, some settings (e.g., $\alpha = 0.7$ for TSPTW50) outperform the default, suggesting that advanced hyperparameter optimization techniques like Bayesian optimization could further enhance performance. We will revise the paper to incorporate these results.
>
> ## **Response 3: The additional runtime overhead**
>
> Thank you for raising concerns about computational efficiency. Below, we clarify the runtime overhead during training and inference.
>
> ### **Training Time**
>
> The training times of the PIP method and the proposed ICO method over 110k epochs **are approximately the same**: for example, around 60 hours for TSPTW50 and about 10 days for TSPTW100. At first glance, fine-tuning $\lambda$ appears computationally intensive. However, in our implementation, we mitigate this by using a fixed number of batches, and in each batch, only the feasible instances are replaced with new ones, as described in lines 251-253. That is, if there is an increasing number of infeasible instances requiring iterative updates in the current batch, the number of newly generated instances in future batches will decrease correspondingly. As a result, the total number of network forward and backward computations remains equivalent to that of the original PIP method.
>
> ### **Inference Time**
>
> Appendix E.5 (Figure 8) provides an anytime performance analysis of hypervolume (HV), infeasibility rate, and optimality gap. The results indicate that ICO consistently outperforms single-$\lambda$ models **throughout the entire iterative process** on HV and gap. The only exception is the infeasibility rate at the initial stages due to the use of a very small initial $\lambda$ value. Importantly, this suggests that even if the additional runtime overhead was reduced or entirely removed, our method would still demonstrate improved performance. This improvement can primarily be attributed to the fact that the adaptively trained policy itself has better capability even without updating $\lambda$ values.
>
> Therefore, we argue that our method's runtime issue is minor. Future work will explore predicting optimal $\lambda$ values to totally eliminate additional overhead. Thank you very much.
>
> ## **Response 4: Add recent studies on adaptive and dynamic methods in constrained combinatorial optimization**
>
> Thank you for valuable comment. We investigated several related works [1, 2, 3] at this aspect and will include discussions about them in the paper. In summary, these related methods proposed many interesting methods for adapting the penalty function, but thay are primarily combined with genetic algorithms instead of neural methods. If there are any omissions, please let us know; we are happy to discuss and compare them.
>
> [1] Adaptive penalty methods for genetic optimization of constrained combinatorial problems, In 1996.
>
> [2] An adaptive penalty scheme for genetic algorithms in structural optimization, In 2004.
>
> [3] Adaptive feasible and infeasible tabu search for weighted vertex coloring, In 2018.

---

### Decision · Program_Chairs · 2025-05-01

**Decision:**

Reject

**Comment:**

The authors propose an instance-level adaptive constraint optimization framework aimed at enhancing feasibility satisfaction in learning-based methods for solving VRPs—an important direction in neural combinatorial optimization. Although the authors made considerable efforts during the rebuttal, the reviewers still highlight that the overall contribution appears incremental relative to existing approaches, especially the PIP. Given the high standards expected at ICML, this point is significant. Consequently, I recommend a weak rejection of the paper in its current form.